# UniVA: Universal Video Agents towards Next-Generation Video Intelligence

## Abstract

While specialized AI models excel at isolated video tasks like generation or understanding, real-world applications demand complex, iterative workflows that combine these capabilities. To bridge this gap, we introduce **UniVA**, a multi-agent framework for universal video intelligence that unifies video understanding, segmentation, editing, and generation in complex workflows. UniVA employs a Plan-and-Act dual-agent architecture: a planner agent decomposes high-level user requests into a sequence of video-processing steps, and executor agents carry out these steps using specialized modular tool servers (for video analysis, generation, editing, object tracking, *etc.*). Through a multi-level memory design (global knowledge, task context, and user-specific memory), UniVA supports long-horizon reasoning and inter-agent communication while maintaining full traceability of each action. This design enables iterative and composite video workflows (*e.g.*, image → video generation → video editing → object segmentation → content composition) that were previously cumbersome to achieve with single-purpose models or monolithic video-language models. We also introduce UniVA-Bench, a benchmark suite of multi-step video tasks spanning understanding, editing, segmentation, and generation, to rigorously evaluate such agentic video systems. Both UniVA and UniVA-Bench are open-sourced to the community, with the aim of catalyzing next-generation video intelligence research. All the code, data, and demos are anonymized at `https://univa-agent.github.io/`

## 1 Introduction

Real-world video applications often require composite, iterative workflows that go beyond any single AI capability (Yu et al., 2023; Maazi et al., 2024; Song et al., 2024). For example, creating a dynamic visual story might begin with an image or text concept, expand into a generated video, then involve editing that video, segmenting key objects, and finally composing multiple elements into a polished scene. Traditionally, accomplishing such a pipeline requires stitching together disparate tools—each specialized for a narrow task—resulting in a brittle, labor-intensive process. The lack of a unified system for reasoning across multiple video tasks and steps has become a critical bottleneck for next-generation video intelligence.

Existing approaches address parts of this challenge but fall short of a unified solution. **Single-task video models** (*e.g.*, dedicated networks for segmentation or video generation) deliver high performance on their specific tasks, yet they operate in isolation and fail to handle multi-step goals without manual coordination. More recently, **unified video-language foundation models** like VILA-U (Wu et al., 2024b) attempt to integrate understanding and generation into one model. These large models learn a broad spectrum of abilities (Fei et al., 2024; Xie et al., 2025; Tan et al., 2025), but they remain monolithic and inflexible – they cannot easily incorporate new tools or modular functions, and leveraging them for complex workflows can be inefficient or impractical. Another emerging direction is to use **LLM-based agents with tool use**. For instance, HuggingGPT employs a language model as a controller to plan tasks and invoke appropriate models/tools in sequence (Shen et al., 2023). Similarly, VideoAgent leverages an LLM with a structured memory and a predefined set of video tools to answer questions on long videos (Fan et al., 2024b). These agent-based systems illustrate the power of planning and tool use (Kugo et al., 2025; Wei et al., 2025). However, HuggingGPT is a generalist framework not specialized for detailed video operations, and VideoAgent focuses mainly on video understanding queries (*e.g.*, Q&A) with limited editing or generation capabilities. To date,

no existing platform fully supports a unified, end-to-end agentic workflow that spans all key facets of video content creation and analysis.

To bridge this gap, we propose **UniVA** (**U**niversal **V**ideo **A**gents), a unified multi-agent video AI platform that enables complex multi-step video creation and manipulation tasks. Technically, UniVA can be depicted by two key characteristics:

- **Highly automated, interactive, proactive user experience:** UniVA is built on a *Plan/Act dual agent architecture*: a planner agent first interprets the user's request and decomposes it into a sequence of actionable steps, and an executor agent (or a team of specialized agents) then carries out each step by invoking the appropriate video tool modules. This separation of planning and acting (in line with recent agent design patterns) allows the system to look ahead and reason about long-horizon goals, while flexibly adapting the plan if intermediate results require changes. On the one hand, with strong planning capabilities, UniVA can autonomously accomplish an entire video production pipeline from a single user query. On the other hand, agents communicate and share information through a *multi-level memory mechanism*: a global memory stores persistent knowledge and facts (*e.g.* general video facts or precomputed embeddings), a task-specific memory retains context and intermediate results for the current workflow, and a user memory keeps track of user preferences or historical interactions. Such memory design ensures that context is maintained throughout the workflow, enabling continuity and avoiding forgetting important details mid-task. In this way, UniVA supports iterative, multi-round interactions, enabling deeply immersive and dynamic creative experiences.

- **Comprehensive, industrial-level video production capabilities:** Built upon the Model Context Protocol (MCP) (Hou et al., 2025), UniVA can seamlessly integrate state-of-the-art video functional modules—either open-source or API-based—in a plug-and-play fashion, where each tool module is implemented as a modular server and the two agents act as the client. The tool hub spans three major categories: video tools (e.g., generation, understanding, editing), non-video tools (e.g., audio or image operations), and non-AI tools (e.g., video cutting). This broad coverage encompasses nearly all functionalities required in the video production process. For example, UniVA supports video generation/transformation from arbitrary conditions, e.g., text, image or video. By combining with cutting-edge external video generation models, UniVA enables cinematic-quality production of long, complex, and narrative-rich videos. Under the MCP framework, the system can also be effortlessly extended to incorporate new tools and capabilities.

To evaluate such systems, we release *UniVA-Bench*, a suite of multi-step video tasks spanning understanding, segmentation, editing, and generation. Tasks are specified as goal cards with gold artifacts (*e.g.*, evidence spans, masks, EDLs) and scored with both task metrics and agentic metrics (plan quality, tool-routing efficiency, memory use, trace completeness). UniVA-Bench is designed to test compositionality, tool swaps, and long-form reasoning—not just per-task accuracy. Code, benchmark, and evaluators are all open-sourced.

In summary, our contributions are threefold: **(1)** We present UniVA, a novel agent-based framework that unifies video tasks in a single open platform. UniVA's plan/act dual-agent architecture with multi-level memory enables it to perform complex, iterative video tasks that were infeasible for previous methods. **(2)** We develop cross-modal and cross-task integration techniques within UniVA, demonstrating how information from one video modality can enhance another – a form of tool synergy that improves outcome quality and coherence. **(3)** We release UniVA-Bench, the first benchmark to assess an agent's competency across a broad range of video tasks and their compositions. Ultimately, UniVA moves the field closer to interactive, next-generation video intelligence that is both highly capable and reproducible.

## 2 RELATED WORK

**Evolution of Video Intelligence.** The field of video intelligence has matured from a collection of specialized tasks—spanning understanding (Tran et al., 2015; Maaz et al., 2024), generation (Ho et al., 2022; Singer et al., 2022), editing (Wu et al., 2023; Liu et al., 2024), and segmentation (Cheng et al., 2023)—into a pursuit for more integrated solutions. In response to the fragmentation of these task-specific systems, a new wave of unified video foundation models has emerged. Models like

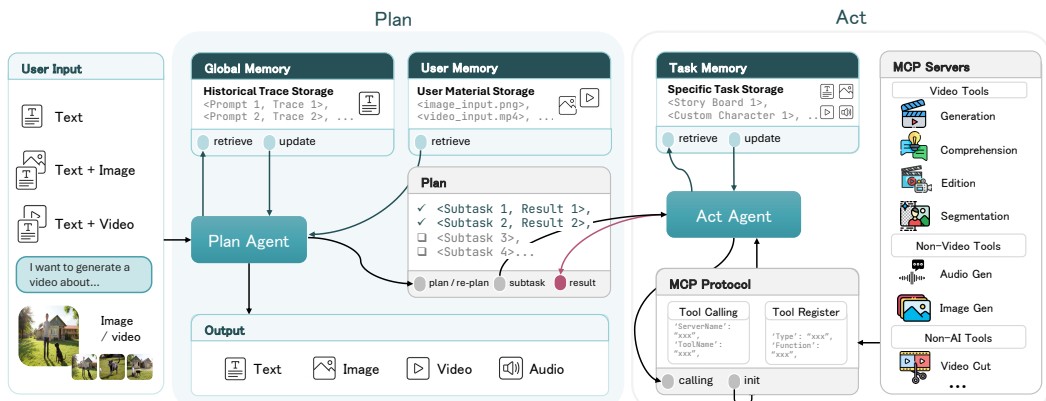

Figure 1: The overall architecture of the **UniVA** system is built on a Plan–Act paradigm. The Plan Agent decomposes user input (text, image, or video) into subtasks by leveraging global memory (historical traces) and user memory (stored materials). The Act Agent then retrieves task-specific memory, executes subtasks via the MCP protocol, and coordinates with external MCP servers, including video, non-video, and non-AI tools. Finally, the system generates versatile, multimodal outputs that span text, image, video, and audio.

Show-o2 and Omni-video aim to jointly train for understanding and generation, while advanced Video-LLMs now incorporate segmentation modules like SAM to enable object-level grounding and reasoning (Xie et al., 2025; Tan et al., 2025; Xiao et al., 2024). Although these unified models represent a significant step forward, they typically rely on static, pre-defined pipelines. This inherent rigidity limits their ability to handle novel task compositions and makes them difficult to extend or maintain. This highlights a critical gap: the need for a framework that moves beyond static integration to enable dynamic, on-the-fly orchestration of heterogeneous modules.

**Agents for Video Intelligence.** Agent-based paradigms have emerged as a promising solution for flexible video intelligence, leveraging planning, interaction, and memory mechanisms (Chen et al., 2024; Yin et al., 2023; Wu et al., 2024a). VIDEOAGENT (Fan et al., 2024a) enhances generative quality with memory augmentation, while other works explore agent planning for long-context reasoning (Wang et al., 2024b) and self-improving generation (Soni et al., 2024). Applications extend to video reasoning (Liu et al., 2025; Shi et al., 2025), editing (Wang et al., 2024a), stylization (Yue et al., 2025), and story generation (Hu et al., 2024). Multi-agent collaborations such as VideoMultiAgents (Kugo et al., 2025) and PREMIND (Wei et al., 2025) further enhance performance, though communication and coordination remain open challenges. Protocols like MCP (Hou et al., 2025) and modular plug-and-play designs offer promising directions. Departing from isolated paradigms, our UniVA framework leverages multi-agent interaction, memory augmentation, and context engineering (Mei et al., 2025) to unify understanding, reasoning, editing, and generation, advancing toward truly universal video agents.

## 3 UniVA

The overall architecture of UniVA, illustrated in Figure 1, is a layered system designed for both extensibility and power. Instead of a monolithic design, UniVA breaks down complex video tasks into a structured workflow. User requests, expressed as natural language or multimodal input, are first processed by the system core. This core consists of two complementary layers: a **Planner** which performs high-level reasoning and decomposes the request into abstract sub-goals, and an **Actor** which grounds these sub-goals into concrete actions by invoking external modules. Their interaction is mediated by a unified **MCP scheduling layer**, which standardizes communication between the Actor and an extensible suite of tools. These tools encompass both domain-specific modules (e.g., understanding, generation, editing, segmentation) and horizontal capabilities (e.g., memory and task control). This layered design achieves a clear separation of concerns: the Planner orchestrates strategy, the Actor manages execution, and the MCP abstracts the heterogeneity of the tool ecosystem. This provides the foundational *breadth* and modularity essential for a comprehensive video intelligence platform. More importantly, the true strength of UniVA lies in its intelligent

control loop, enabled by the synergistic interplay between the dual agents and multilevel memory, which unlocks the system's depth and overall effectiveness.

## 3.1 PLAN–ACT DUAL AGENT ARCHITECTURE

As the core of UniVA, the *Plan–Act dual-agent architecture* separates high-level reasoning from low-level execution while maintaining a tight feedback loop between the two:

- The **Planner** is responsible for interpreting the user request and breaking it down into a series of sub-goals. For example, given "make a cartoon video of my dog", the Planner may decompose it into: (1) retrieve images of the dog, (2) generate a cartoon-style video, (3) edit the background, and (4) compose audio. The Planner maintains high-level reasoning, determines the task order, and dynamically adapts the plan when intermediate results suggest revisions.
- The **Actor** is responsible for execution. It receives each sub-goal from the Planner, selects the appropriate tool through the MCP interface, fills in required arguments (e.g., video clip, mask, prompt), and executes the call. Once a tool finishes, the Actor collects the output and sends it back to the Planner. This separation keeps the Planner lightweight and strategic, while the Actor focuses on using the tool reliably and efficiently.

Together, the Planner and Actor form an iterative control loop: the Planner decides what to do next, the Actor executes the decision, and memory modules record intermediate outputs. This design enables UniVA to handle complex, multi-step video tasks in a structured yet flexible manner.

## 3.2 MEMORY MECHANISM

A key challenge in agentic video systems is to maintain context across long and multi-step workflows. As presented in Figure 2, UniVA addresses this with a *three-level memory mechanism* that augments the Planner–Actor loop:

1) **Global Memory.** Stores persistent knowledge and reusable resources, such as precomputed embeddings, generic video facts, or tool usage statistics. This memory provides background context and supports cross-task generalization.
2) **Task Memory.** Maintains intermediate artifacts, tool outputs, and execution traces within the current workflow. It ensures continuity across multiple steps, allowing later sub-goals to reuse results (e.g., segmentation masks or captions) without redundant computation. Task memory also enables traceability, making the entire workflow transparent and reproducible.
3) **User Memory.** Captures user-specific preferences and historical interactions, such as favored styles, recurring edit patterns, or personalized constraints. This enables adaptive behaviors, e.g., automatically applying a user's preferred editing style in future tasks.

Together, these memory levels provide both persistence and adaptability, ensuring that UniVA can efficiently manage context, reuse knowledge, and tailor its outputs to diverse user needs.

## 3.3 MCP SERVERS

The MCP server module acts as a unified gateway between the Actor and a collection of distinct tool servers. The server maintains a registry of available functions, validates and executes calls through a standardized API, and records outputs for traceability. This design means that adding or replacing a capability only requires registering it on the server, while the Planner and Actor remain unchanged, making the system modular and extensible.

UniVA is equipped with an extensive and diverse toolset integrated via this MCP layer. To balance flexibility with reliability, we classify each function as either an [Atom] or a [Workflow]:

- **[Atom]**: A fundamental, single-purpose operation, such as generating an image from a text prompt.

- **[Workflow]**: A higher-level, pre-composed function that orchestrates multiple atomic tools to complete a common multi-step task, such as generating an entire story video from a single command.

This dual approach provides the Planner with both the versatility to creatively combine atomic tools for novel problems and the stability of using robust workflows for high-stakes production. A com-

Figure 2: Memory-augmented framework for video generation. Global and user memories provide context to the Plan Agent, while task memory coordinates tool calling, storyboard creation, and the overall video generation process.

prehensive taxonomy of our entire toolset, spanning video, audio, and image modalities, is provided in Appendix C.3 for a full overview of UniVA's capabilities.

The unified architecture of UniVA is designed to accommodate a wide spectrum of creative workflows, from fine-grained, iterative refinement to fully autonomous, end-to-end execution. For users who prefer a hands-on approach, the framework's breadth of modular tools can be orchestrated in an interactive, multi-round dialogue, allowing for precise, step-by-step control. Conversely, for users who desire a more streamlined process, UniVA's depth allows it to autonomously handle a single, high-level instruction. By leveraging the synergistic interplay of its Planner, Actor, and memory systems, the agent can independently decompose a simple prompt into a complex, multi-shot narrative workflow. We provide visual walkthroughs in Appendix C.2 to further illustrate these capabilities.

## 4 UNIVA-BENCH

Real-world video creation is inherently an iterative, multi-stage process, yet existing benchmarks largely evaluate video intelligence through isolated, single-model tasks. This overlooks critical agentic capabilities, such as planning, memory utilization, and tool orchestration. To bridge this gap, we introduce **UniVA-Bench**, an agent-oriented benchmark that shifts the focus to end-to-end, tool-augmented workflows, aligning evaluation with the complex demands of practical video agents.

### 4.1 TASKS

We define the task taxonomy in UniVA-Bench as follows, detailed curation processes are provided in Appendix D.2:

- **Understanding (Long-Video QA).** This task is designed to target both aesthetics- and semantics-oriented questions for long videos, encompassing shot transitions, visual style, and narrative comprehension in addition to standard entity and action semantics. Unlike prior settings, where each QA pair is tied to a single short clip, our task demands answering multiple interdependent questions grounded in a single long-form video.
- **Generation.** Agents are evaluated on diverse real-world video generation tasks, categorized into three subtypes: *1) LongText2Video*, handling long or noisy prompts that necessitate storyboard-first planning; *2) Image/Entities2Video*, using 1–3 reference images to enforce identity preservation and cross-scene coherence; *3) Video2Video*, conditioning on a source video while ensuring referential stability for persons and objects.

- **Editing (Long Video).** This task is defined to involve multi-step edits such as cross-shot replacement, attribute modification, and style transfer, while maintaining narrative integrity and referential consistency. Effective completion requires reasoning in combination with tool invocation (e.g., ref-seg → inpaint/compose → merge).
- **Segmentation (Long Video).** Designed for long clips with multiple entities and frequent occlusions, this task evaluates temporal consistency and robustness in detecting and segmenting shot boundaries.
- **Agentic probing sets.** The sets are designed to probe agentic capabilities explicitly, including: (1) a 50-instance *storyboard-to-user-intent planning set* to compare Single-Agent versus Plan-Act, and (2) a *pipeline-based task set* with expert references, used to evaluate Weighted Plan Edit Distance (wPED), Dependency Coverage (DepCov), and Re-planning Quality (ReplanQ) under injected failures. In addition, memory-oriented analyses consider three dimensions: *trace memory* (e.g., historical trajectories), *user memory* (personal preferences), and *task memory* (e.g., storyboards).

## 4.2 EVALUATION PROTOCOL

To evaluate agent performance on UniVA-Bench, we employ a comprehensive suite of metrics targeting three key areas: (1) Task-specific Quality, using established metrics like CLIP Score for command following and DINO Score for subject consistency; (2) Overall User Preference, captured via pairwise judgments from a powerful MLLM-as-a-Judge; and (3) Agentic Planning Capabilities, assessed using our novel, specialized metrics (wPED, DepCov, and ReplanQ) that measure plan quality, logical correctness, and recovery robustness. The detailed definitions and calculation methods for all metrics are provided in Appendix D.3.

## 5 EXPERIMENTS

To comprehensively evaluate our system's capabilities in realistic, end-to-end workflows, we conduct all experiments on UniVA-Bench, an agent-oriented benchmark we introduce in this work. Our experimental design is guided by two central hypotheses: **1)** that a unified agentic architecture, where functional modules like understanding and generation are deeply integrated, provides a significant performance advantage over isolated, end-to-end models; and **2)** that the combination of a dual-agent Plan-Act framework and a multi-component memory system is essential for achieving the robust planning and persistent context required for complex video tasks.

## 5.1 PERFORMANCE OF FUNCTIONAL MODULES

**Generation.** In the generation scenarios, we benchmark UniVA against three representative end-to-end models: LTX-Video (HaCohen et al., 2024), Wan (Wan et al., 2025), and Seedance (Gao et al., 2025). We evaluate the results using CLIP Score (prompt following), DINO Score (subject consistency), and preference ratings from an MLLM-as-a-Judge, following the UniVA-Bench specification. The results are shown in Table 1. For **LongText2Video**, UniVA achieves the highest CLIP score of 0.2814 and the MLLM Judge score 3.333, which is directly attributable to its agentic framework. Unlike end-to-end models, UniVA's Planner first parses the noisy, long-term text to distill the core user intent into an optimal prompt, overcoming a common shortage of traditional end-to-end models. On **Entities2Video**, while specialized models like Seedance show strong performance in subject consistency (DINO Score), UniVA remains competitive. This highlights a current trade-off where our agent prioritizes overall instruction complexity and narrative coherence, a direction for future optimization. Regarding **Video2Video**, although UniVA does not lead in automated metrics such as the CLIP Score or DINO Score, it achieves a commanding MLLM Judge score of 4.068. This apparent discrepancy shows that UniVA's planner excels at interpreting and executing complex instructions (e.g., 'modify the storyline while preserving the style'). This often requires a correct understanding of the original video, and then providing a concise prompt to generate a new video.

**Understanding.** For the understanding task, we compare UniVA against several leading Large Multimodal Models, including GPT-4o (OpenAI et al., 2024), Gemini 2.5 Pro (Google, 2023), InternVL3-38B (Zhu et al., 2025), and Qwen2.5-VL-72B (Bai et al., 2025). Performance is measured by QA accuracy. As shown in Table 2a, UniVA achieves the highest accuracy of 0.76, outper-

Table 1: Comparison across LongText2Video, Entities2Video and Video2Video.

| Method | LongText2Video | | | Entities2Video | | | Video2Video | | |
|---|---|---|---|---|---|---|---|---|---|
| | CLIP Score | DINO Score | MLLM Judge | CLIP Score | DINO Score | MLLM Judge | CLIP Score | DINO Score | MLLM Judge |
| LTX-Video | 0.2161 | **0.9392** | 1.125 | 0.2210 | 0.8452 | 1.281 | 0.2263 | **0.9943** | 2.123 |
| Wan | 0.2028 | 0.6779 | 3.183 | **0.3106** | 0.7043 | 1.650 | 0.2632 | 0.9188 | 2.034 |
| Seedance | 0.2157 | 0.8836 | 2.650 | 0.3039 | **0.8800** | **2.700** | **0.2684** | 0.9518 | 2.621 |
| **UniVA** | **0.2814** | 0.9026 | **3.333** | 0.2868 | 0.8796 | 1.789 | 0.2620 | 0.8939 | **4.068** |

Table 2: Comparison of three long video tasks: Understanding, Editing, and Segmentation.

(a) LongVideo Understanding

| Method | Acc |
|---|---|
| GPT-4o | 0.52 |
| Gemini 2.5 Pro | 0.65 |
| InternVL3-38B | 0.75 |
| Qwen2.5-VL-72B | 0.74 |
| **UniVA** | **0.76** |

(b) Long Video Editing

| Method | Editing | | |
|---|---|---|---|
| | CLIP | DINO | MLLM |
| Aleph | 0.2258 | 0.6808 | 3.484 |
| **UniVA** | **0.2280** | **0.7488** | **3.635** |

(c) Long Video Segmentation

| Method | Segmentation | | |
|---|---|---|---|
| | J | F | J&F |
| SA2VA | 0.2076 | 0.0972 | 0.1524 |
| **UniVA** | **0.3254** | **0.1680** | **0.2467** |

forming the isolated models. These results highlight the benefit of UniVA's agentic design, where explicit task decomposition and iterative reasoning allow the system to maintain context and handle complex, multi-question queries more effectively than single-pass inference models.

**Editing.** For long video editing, we benchmark UniVA against Runway Aleph (run, 2025), a strong baseline for video editing tasks. Evaluation metrics include CLIP Score, DINO Score, and MLLM preference. As demonstrated in Table 2b, it can be seen that in a traditional set-up, an editing model would be disconnected from a deep and continuous understanding of the video. UniVA bridges this gap by first leveraging the integrated Understanding module through the Probing tool to establish a persistent semantic context, enabling the agent to accurately ground editing objects across long-term, multi-shot video and to execute edits in a coherent and context-aware manner.

**Segmentation.** In the challenging long video segmentation task, we use SA2VA (Yuan et al., 2025a) as our primary baseline. We report the J-mean, F-mean, and J&F-mean scores. In Table 2c, UniVA achieves the best scores on all metrics. This is because UniVA can query the co-located Understanding module to resolve ambiguities that are impossible to solve at the pixel level. For instance, when an object is occluded, the agent can ask the Probing tool: "Based on the narrative context, is the object reappearing at timestamp X the same 'blue car' from timestamp Y?" This ability to dynamically leverage a powerful understanding module to inform a perception task like segmentation is a unique benefit of our integrated design.

> These 4 experiments demonstrate that a unified agentic architecture is critical for advancing video intelligence. The superior performance of UniVA is not merely due to the quality of its individual modules but stems from the tight coupling and dynamic interplay between them.

## 5.2 AGENTIC SYSTEM PROBING

### 5.2.1 PLANNING CAPABILITY

**Analysis:** To select the optimal Planner for our framework, we evaluated three leading LLMs (Figure 3). Claude-Sonnet-4 demonstrated superior performance in DepCov and ReplanQ. Since correctly identifying task dependencies and robustly recovering from failures are paramount for a reliable agent, we selected Claude-Sonnet-4 as the Planner for all subsequent experiments.

In Figure 4, Success Rate is the percentage of test cases where the agent produced a structurally valid plan (*i.e.*, $wPED > 0$)—measuring the agent's ability to avoid catastrophic failures, such as generating an empty or malformed output. It more than doubles the Success Rate (45.0% vs. 20.0%), indicating a much lower rate of catastrophic failures. Furthermore, the quality of its successful plans is also over twice as high, reflected in a wPED score of 0.117 versus 0.050. This confirms that the explicit planning stage can not only output valid plans but also high-quality plans.

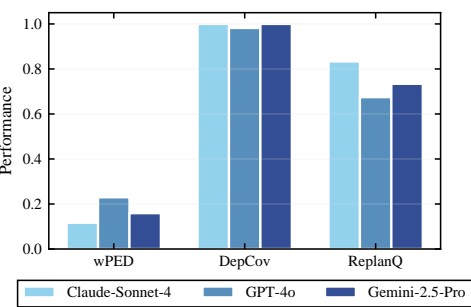

Figure 3: Performance of Planner LLMs.

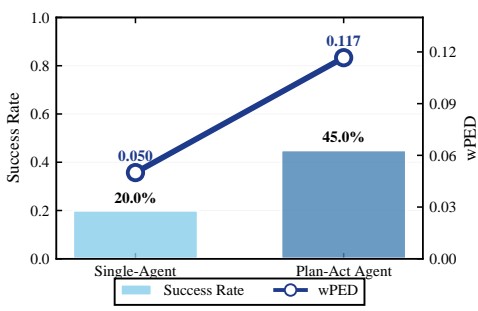

Figure 4: Framework comparison

### 5.2.2 MEMORY CAPABILITY

We then analyze the distinct contributions of our three memory modules. To isolate their effects, we designed specific experimental probes: (i) Global Memory was tested by providing the agent with a set of trajectories from an expert planning dataset; (ii) User Memory was evaluated in the Entities2Video task, where the agent could retrieve user-provided reference images via a RAG mechanism; and (iii) Task Memory was assessed in the LongText2Video task by comparing the performance of generating with and without a storyboard.

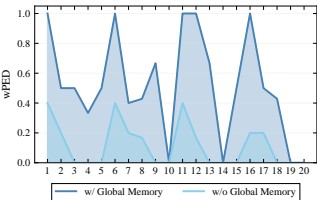

Figure 5: Trace Memory.

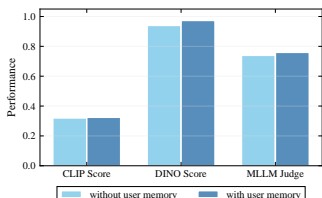

Figure 6: User Memory.

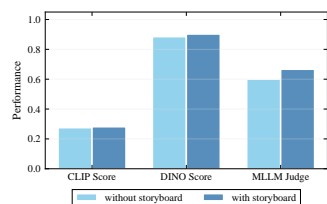

Figure 7: Task Memory.

**Analysis:** In Figure 5, across most cases, the agent with global memory achieves a higher wPED score than without global memory. This indicates that by drawing on past trajectories, the agent becomes better at aligning its generated plans with expert-preferred structures. And most strikingly, global memory prevents catastrophic planning failures. In numerous instances (e.g., turns 3-5, 8-10, 14, and 18-20), the agent without global memory completely fails to produce a viable plan, resulting in a wPED score of zero. However, an agent with global memory not only succeeds but often produces high-quality plans. Figure 6 shows that with the user memory, the agent can better understand the user's indications, such as when the user refers to a cat, the user memory can enable the agent to locate the cat image from the user's materials, making the generated content more aligned with user intent. Utilizing storyboards as task memory (Figure 7) provided a substantial boost across all quality metrics. This demonstrates that maintaining an intermediate representation of the creative goal is essential for ensuring semantic coherence and cross-shot consistency in the final video, directly validating the storyboard's role in our agent's workflow.

> In summary, our dual Plan-Act Agent framework improves the ability to process complex tasks. Additionally, three memory mechanisms enable the agent to build a persistent context, making it more robust, facilitating better user intent understanding, and ensuring more consistent generation of videos.

### 5.3 HUMAN EVALUATION

To complement our automated evaluations and validate the MLLM-as-a-Judge, we conducted a formal human evaluation study. The primary goal is to determine if the MLLM-as-a-Judge corresponds with the subjective preferences of human annotators. We focus on the video generation tasks (Long-Text2Video, Image2Video, and Video2Video). We collected generated video results from both our

UniVA system and the baseline models for each task. Annotators were asked to judge each video based on a set of criteria identical to those used for the MLLM-as-a-Judge.

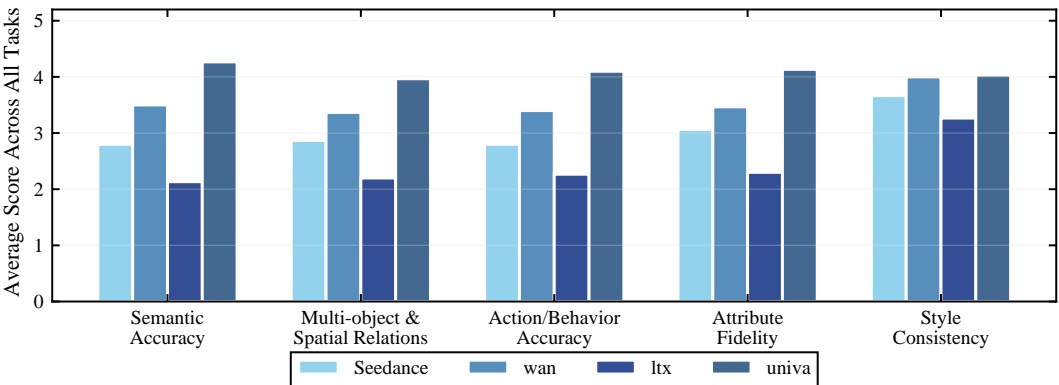

Figure 8: Results from the human evaluation study on video generation tasks.

UniVA (darkest blue bar) emerges as the clear leader, achieving the highest human preference scores in four out of the five evaluated dimensions. This strong human preference aligns with the patterns observed in our automated metrics, confirming that our MLLM judge is a reliable proxy for genuine human perception.

### 5.4 QUALITATIVE CASE STUDIES

To provide a more intuitive understanding of these quantitative results, we present a series of qualitative case studies in Appendix F. These examples visualize how UniVA's unique capabilities in planning and synergy lead to superior outcomes in complex narrative scenarios.

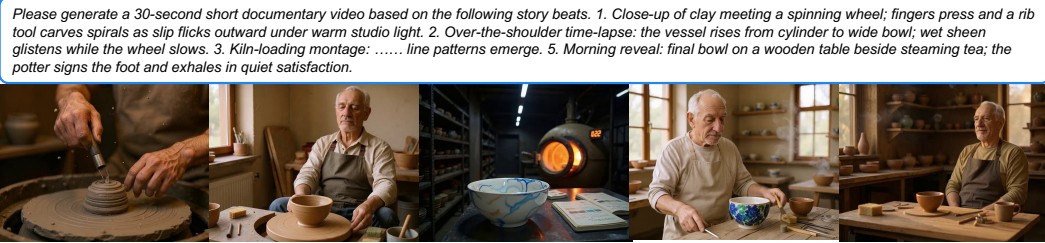

Figure 9: UniVA accurately generates sequential process of pottery making, demonstrating strong temporal consistency and object persistence as the bowl evolves from clay to finished product.

## 6 CONCLUSION

In this work, we introduced UniVA, a unified agentic framework designed to tackle the next frontier of video intelligence. We argued that progress in the video domain requires a paradigm shift from developing isolated, single-task models to creating integrated systems capable of complex, collaborative workflows. To this end, our primary contributions were the development of the powerful and extensible UniVA platform, the demonstration of its emergent synergistic capabilities, and the release of UniVA-Bench to rigorously measure such advancements. Our experiments validate UniVA's breadth, demonstrating competitive performance across a wide array of video tasks. More profoundly, we reveal its depth: through "Agentic Synergy" enabled by the dynamic management of information flow between tools, UniVA solves complex consistency problems intractable for siloed models. This confirms that UniVA is not merely a collection of tools, but an engine for generating emergent intelligence. We hope that UniVA and UniVA-Bench will inspire future video intelligence research into this new generation of integrated, synergistic AI systems.

ETHICS STATEMENT

We have developed UniVA and UniVA-Bench with a strong commitment to the ICLR Code of Ethics. We have carefully considered the potential ethical implications of our work and detail our considerations regarding key areas below.

**Potential for Misuse of Generative Technology**   We acknowledge that powerful video generation and editing technologies, such as UniVA, have the potential for dual use. While our primary goal is to create a tool that empowers creative professionals and democratizes video production, this technology could be misused to create convincing synthetic media for malicious purposes, including misinformation and harassment. The current version of UniVA does not incorporate explicit safeguards against the generation of harmful, biased, or copyrighted content. We believe that the development of robust detection methods, watermarking techniques, and responsible deployment practices is critical research directions for the entire community, and we are committed to contributing to these efforts in future work.

**Data Licensing, Bias, and Curation**   Our research utilizes several publicly available datasets (e.g., Video-MME, SF20k, DAVIS2017), and we have adhered to their respective licenses and terms of use. We also acknowledge that the large pre-trained models used within UniVA (e.g., Claude, Qwen2.5-VL) may have inherited societal biases from their training data, which could be reflected in the generated outputs. Furthermore, for the curation of UniVA-Bench, we employed LLMs to generate certain prompts and question-answer pairs. This process may introduce biases inherent in the LLMs themselves. To mitigate this, all machine-generated data underwent a rigorous manual review and refinement process by the authors to ensure quality, diversity, and alignment with the benchmark's goals. We plan to release UniVA-Bench under a license that encourages responsible research use.

**Human Subjects in User Study**   Our human evaluation study involved human annotators. All participation was voluntary, and the purpose of the study was clearly explained to all participants beforehand. To protect privacy, all data collected was fully anonymized, and no personally identifiable information was stored. The task involved rating generated videos for quality and relevance, which we assess to be a low-risk activity.

**Research Integrity**   We are committed to the integrity of our research. The experimental results are reported transparently, and we have made an effort to fairly compare our system against relevant baselines. In the appendix, we provide detailed configurations for our system and all baselines, as well as precise definitions for our evaluation metrics, to ensure the reproducibility of our findings. A statement on our use of LLMs in the research and writing process is also included in the appendix.

REPRODUCIBILITY STATEMENT

We are committed to ensuring the reproducibility of our research. To this end, we provide a comprehensive set of resources and detailed documentation, distributed across our main paper, the appendix.

**Codebase.**   We will release the complete codebase for the UniVA framework, including the implementation of the dual-agent control system, the multi-level memory, and the MCP-based tool integration layer.

**Benchmark and Datasets.**   Our novel benchmark, UniVA-Bench, including all curated prompts, evaluation data, and reference plans, will be publicly released. A detailed description of the data curation process for each sub-task is provided in Appendix D.

**Experimental and Evaluation Details.**   To facilitate the replication of our experimental results, we have provided extensive details in the appendix. Appendix E contains the precise configurations for our UniVA system and all baseline models, including key hyperparameters. Appendix D offers the exact definitions and mathematical formulas for all our evaluation metrics, along with the prompt template used for the MLLM-as-a-Judge.

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

APPENDIX INDEX

## A    STATEMENT ON LLM USAGE

As detailed in Section D, we utilized Large Language Models (LLMs) to programmatically assist in the data curation process for our UniVA-Bench. Specifically, LLMs were used to generate multiple-choice question-answer pairs for the Long-Video QA task, and to create noisy prompts for the LongText2Video task. It is important to note that all machine-generated data underwent a rigorous manual review, filtering, and refinement process by the authors to ensure its quality, relevance, and alignment with the benchmark's objectives. The authors take full responsibility for all content presented in this work.

## B    HIGHLIGHT OF UNIVA

To provide a concise overview of UniVA's overall capabilities and unique characteristics, we summarize its key highlights below. These can be categorized into two primary value propositions: a revolutionary user experience and comprehensive, industrial-grade production power.

**1) Highly automated, interactive, proactive user experience.**    UniVA is architected to transform the traditionally complex and labor-intensive video creation process into a fluid, intuitive, and collaborative dialogue. Through its decoupled, modular design and intelligent dual-agent control system, UniVA delivers a user experience that is proactive, iterative, and highly automated.

- **Proactive Planning.**    Instead of passively waiting for precise, step-by-step commands, UniVA's Planner proactively analyzes high-level user goals to formulate robust, multi-step execution plans.
- **Seamless Iteration.**    The combination of the Actor's precise execution and the Task Memory's stateful tracking enables a truly iterative workflow. Users can incrementally build upon previous results, make fine-grained adjustments.
- **Extensible Platform.**    MCP-based architecture ensures that UniVA is not a static system. It is designed to be an extensible platform where new, state-of-the-art models and capabilities (e.g., emerging audio or new stylization models) can be seamlessly integrated as new tools.

**2) Comprehensive, industrial-level video production capabilities.**    Beyond its user-centric design, UniVA is a powerful production engine built for versatility and quality. By integrating a comprehensive suite of state-of-the-art models within a unified framework, UniVA offers an unprecedented breadth of capabilities and delivers results that meet the standards of professional and industrial applications.

- **Any-conditioned Video Gen.**    UniVA breaks the rigid boundaries of traditional generation models. It can synthesize video from virtually any combination of inputs: long and complex text, reference images, existing video clips, character likenesses, and more. This flexibility allows creators to work with the materials they have, rather than being constrained by the tool's limitations.
- **Coherent Long-Form Video.**    UniVA excels at producing long-form videos with high temporal and semantic coherence. It can maintain character identity, object persistence, and stylistic consistency across multiple scenes.

Figure 10 provides a visual summary of UNIVA's highlights.

## C    DETAILED METHODOLOGY

Since UniVA integrates multiple functionalities within a large-scale system, it is essential to clarify its design philosophy. In this section, we present the guiding principles and functional workflow that underpin the framework, providing a comprehensive view of our design.

### C.1    PRINCIPLES

Here we provide a set of design principles that guide the construction of our UniVA framework.

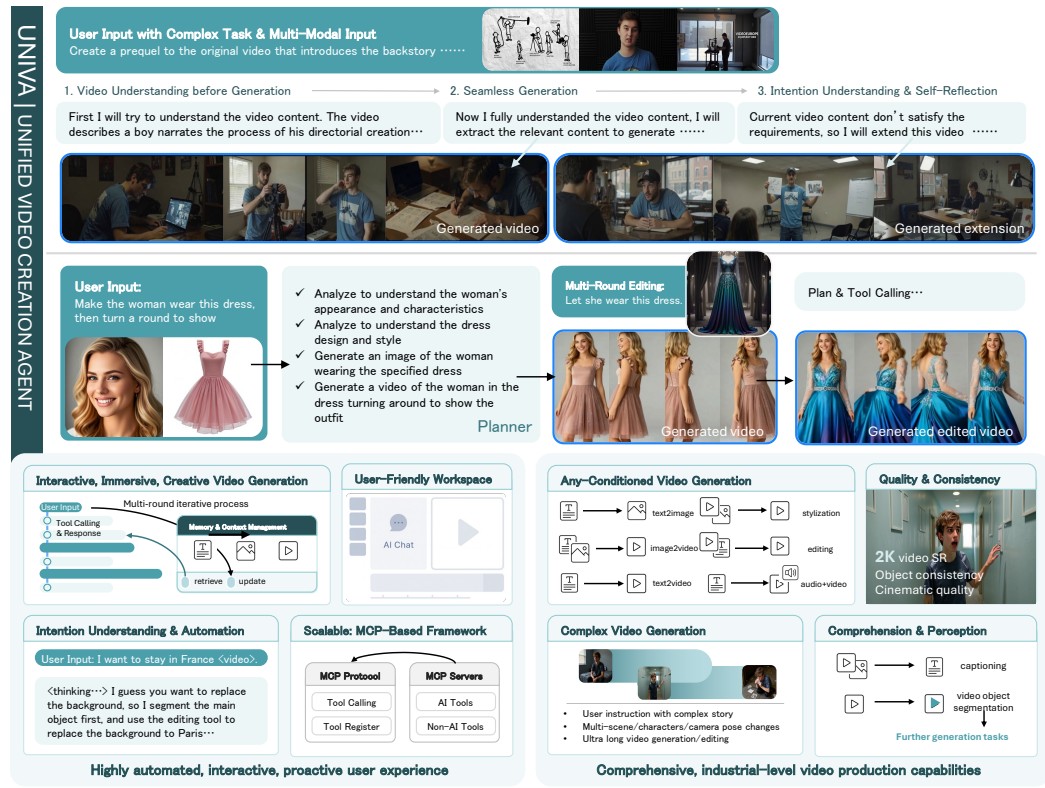

Figure 10: A visual overview of highlights of the proposed UNIVA.

**1. Unified & Modular Architecture.** A comprehensive and extensible system requires a modular architecture. In UniVA, all capabilities—from SOTA generation models to simple non-AI tools—are integrated as decoupled functional modules. These modules are invoked via a unified Model Context Protocol (MCP), allowing them to be updated or replaced in a plug-and-play fashion. This principle is the foundation for the system's industrial-level production capabilities and ensures it can consistently deliver cinematic-quality output by leveraging the best available tools.

**2. Separation of Plan & Act for Complex Workflows.** At the core of the agent's operation is a strict Plan-Act separation, which realizes the dual agent architecture described previously. A Planner agent interprets high-level user intent and decomposes it into a logical sequence of steps. An Executor agent then carries out each step by invoking the appropriate tools. This separation is crucial for managing long-horizon tasks and allows the system to robustly handle complex, multi-step video production pipelines.

**3. Proactive, Goal-Oriented Autonomy.** Crucially, the Planner is more than a passive task decomposer; it is designed for a high degree of automation and proactive behavior. The agent actively evaluates intermediate results against the inferred user goal. If an output does not align with the objective—as shown in the teaser, where the agent decides a video is insufficient—it initiates self-reflection to flexibly adapt its plan. This ability to autonomously course-correct is the key to accomplishing an entire production pipeline from a single user query.

**4. Hierarchical Memory for Immersive Interaction.** The framework's multi-level memory mechanism is what enables iterative, multi-round interactions and deeply immersive creative experiences. This hierarchy consists of global memory for persistent knowledge, task-specific memory to maintain context for the current workflow, and user memory to track preferences. This design ensures contextual continuity, allowing users to refine and build upon their creations over extended interactions.

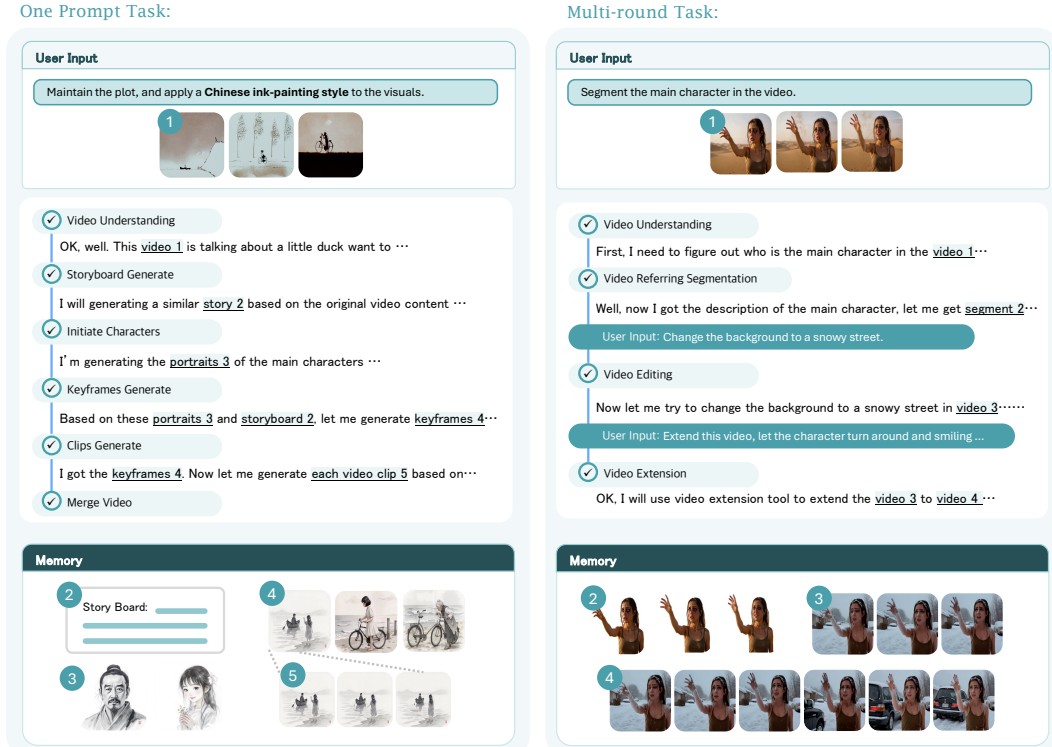

Figure 11: An iterative tool calling for video generation. Left: one-prompt task applies a global ink-painting style. Right: multi-round task incrementally edits via segmentation, background change, and extension, demonstrating representative functions.

**5. Composition of Atomic Operations into Robust Workflows.**   To effectively handle the composite and iterative workflows mentioned earlier, the framework strikes a balance between flexibility and reliability. All complex functionalities are built upon a set of fine-grained atomic operations. The Planner can creatively combine these atomic operations to solve novel problems. For common, high-stakes tasks, these operations are organized into pre-defined workflow patterns to ensure robust and predictable execution. This dual approach provides the system with both the versatility for creative exploration and the stability required for industrial-grade production.

## C.2   END-TO-END WORKFLOW DEMONSTRATION

Figure 11 showcases how UniVA's components work in synergy, revealing both its depth in handling complex, autonomous tasks and its breadth in supporting interactive, multi-tool creation.

The one-prompt task (left panel) exemplifies the system's depth. Faced with a complex command, the *Plan-Act agent* autonomously decomposes the goal and orchestrates a sequence of tools via the *MCP Servers*. By managing the information flow through the *Memory Mechanism*, it effectively connects different capabilities, such as using an *understanding* tool to empower a *generation* tool. This enables the agent to collaboratively use multiple tools to achieve a sophisticated goal in a single pass.

Conversely, the multi-round task (right panel) highlights the system's breadth. It provides a powerful platform with a wide array of tools that users can flexibly combine through iterative interaction. Each command triggers a new Plan-Act cycle, where the agent leverages context from the *Memory Mechanism* (e.g., a segmentation mask) to execute the next step. This demonstrates how our architecture supports a flexible, stateful, and collaborative creative process.

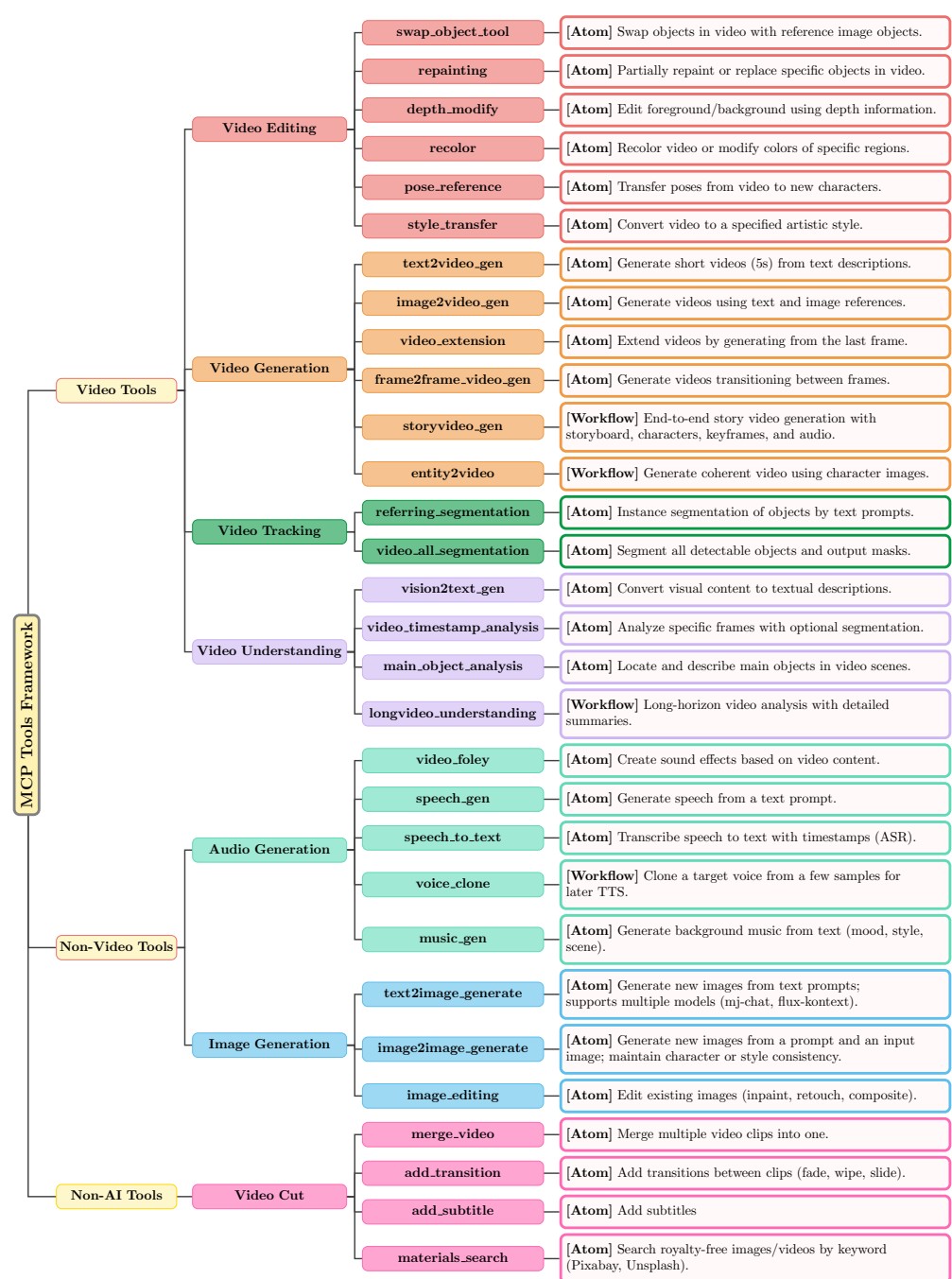

Figure 12: A three-level taxonomy of MCP tools: modules (level-1), tools (level-2), and leaf boxes summarizing name, type, and functionality.

## C.3 FUNCTION WALKTHROUGH

UniVA is equipped with an extensive, modular toolset integrated via the Model Context Protocol (MCP). This "plug-and-play" architecture enables the agent framework to invoke a diverse range of specialized functions. As shown in Figure 12, these tools are organized into three main categories: Video Tools, Non-Video Tools, and Non-AI Tools.

Each function is classified as either an `[Atom]` or a `[Workflow]`:

- **[Atom]**: A fundamental, single-purpose operation, such as generating an image from text.

- **[Workflow]**: A higher-level function that composes multiple atomic tools to complete a multi-step task, such as generating an entire story video.

### C.3.1 VIDEO TOOLS

This core category encompasses functionalities for video creation, modification, and analysis.

**Video Editing.** Provides granular control over the visual elements within a video.

- **swap_object_tool** [Atom]: Swaps objects in a video with those from a reference image.

- **repainting** [Atom]: Repaints or replaces specific objects within a video.

- **depth_modify** [Atom]: Edits the foreground or background of a video using depth information.

- **recolor** [Atom]: Recolors an entire video or modifies the colors of specific regions.

- **pose_reference** [Atom]: Transfers poses and movements from a source video character to a new one.

- **style_transfer** [Atom]: Applies a specified artistic style to a video.

**Video Generation.** Creates new video content from various inputs.

- **text2video_gen** [Atom]: Generates short videos (approx. 5s) from text descriptions.

- **image2video_gen** [Atom]: Generates videos from a text prompt and an image reference.

- **video_extension** [Atom]: Extends a video by generating subsequent frames.

- **frame2frame_video_gen** [Atom]: Generates a video transitioning between a start and end frame.

- **storyvideo_gen** [Workflow]: End-to-end story video generation, including storyboard, characters, keyframes, and audio.

- **entity2video** [Workflow]: Generates a coherent video using a set of character images.

**Video Tracking.** Identifies and isolates objects or regions within a video.

- **referring_segmentation** [Atom]: Segments video objects based on text prompts.

- **video_all_segmentation** [Atom]: Segments all detectable objects in a video and outputs their masks.

**Video Understanding.** Analyzes and extracts semantic information from video.

- **vision2text_gen** [Atom]: Generates a textual description of a video's visual content.

- **video_timestamp_analysis** [Atom]: Analyzes specific frames, with optional segmentation for focused analysis.

- **main_object_analysis** [Atom]: Locates and describes the main objects in video scenes.

- **longvideo_understanding** [Workflow]: Analyzes long videos to provide detailed summaries and insights.

### C.3.2 NON-VIDEO TOOLS

This category includes functionalities for audio and image creation, editing, and synchronization.

**Audio Generation.**

- **video_foley** [Atom]: Create and sync sound effects (foley) to visual events.
- **speech_gen** [Atom]: Generate speech from a text prompt.
- **speech_to_text** [Atom]: Transcribe speech to text with timestamps (ASR).
- **voice_clone** [Workflow]: Clone a target voice from a few samples for later TTS.
- **music_gen** [Atom]: Generate background music from text (mood, style, scene).

**Image Generation.**

- **text2image_generate** [Atom]: Generate images from text prompts (e.g., `mj-chat`, `flux-kontext`).
- **image2image_generate** [Atom]: Generate a new image from a prompt conditioned on an input image for style/identity consistency.
- **image_editing** [Atom]: Edit existing images (inpaint, retouch, composite).

### C.3.3 NON-AI TOOLS

This category provides deterministic utilities for cutting, merging, and augmenting video materials.

**Video Cut.**

- **merge_video** [Atom]: Merge multiple clips into a single sequence.
- **add_transition** [Atom]: Add transitions between clips (fade, wipe, slide).
- **add_subtitle** [Atom]: Add subtitles.
- **materials_search** [Atom]: Search royalty-free images/videos by keyword (e.g., Pixabay, Unsplash).

## D UNIVA-BENCH

### D.1 BENCHMARK DEFINITION

**Motivation.** Video intelligence in practice is an iterative, multi-stage creation process where users interleave understanding, generation, editing, segmentation, and audio/asset composition within a single workflow. However, most existing benchmarks largely isolate single tasks and single models, which underestimates the difficulty of long-horizon, multi-step video production and the need for explicit planning, memory, and tool orchestration. Therefore, we introduce a unified agent-oriented benchmark that covers four core video capabilities under multiple real-world conditions: (i) Understanding, (ii) Generation, (iii) Editing, (iv) Segmentation.

UniVA-Bench shifts the focus from isolated single-model tasks to end-to-end, tool-augmented video intelligence, aligning evaluation with real user workflows and the requirements of practical video agents.

**Tracks.** The benchmark consists of two complementary tracks:

- **Functional Modules:** task performance across Understanding, Generation (LongText2Video, Image/Entities2Video, Video2Video), Editing (long video edits with cross-shot consistency), and Segmentation (long video segmentation with multi-entity occlusion).
- **Agentic Probing:** plan quality, dependency satisfaction, and re-planning robustness using structured plan-level metrics; analysis of memory usage (trace, user, task/storyboard) and its downstream impact.

**What it evaluates.** (1) Task competence on long-form video; (2) Multi-tool coordination and referential stability across shots/entities; (3) Planning structure and dependency coverage; (4) Effects of memory on controllability, predictability, and recovery.

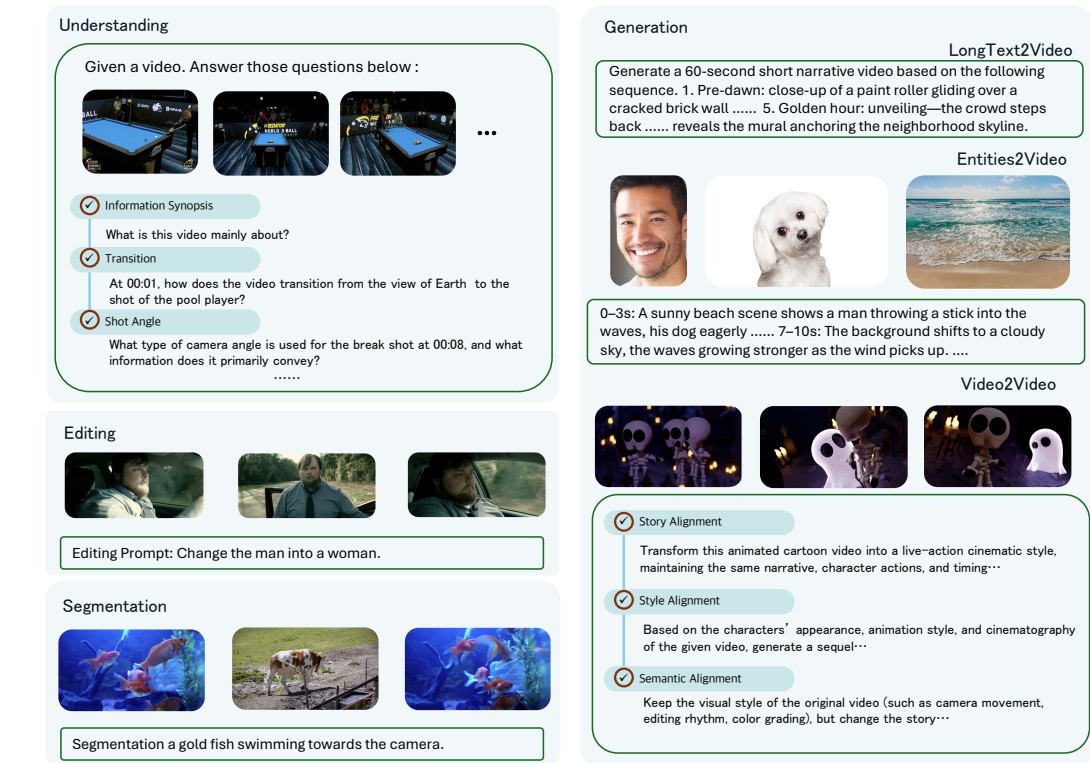

Figure 13: Benchmark demo cases.

## D.2 DATA CURATION

**Understanding (Long-Video QA).** We randomly sampled 10 videos from Video-MME Fu et al. (2025) and used Gemini 2.5 Pro to generate multiple-choice QA pairs based on the perspectives shown in Table 3. The task specifies that each video corresponds to 10 questions, and all answers must be provided within a single inference.

**Generation.** In the data curation stage, for the LongText2Video task, we first use GPT to generate a clear storyboard, then rewrite it into long and noisy prompts. For the Image/Entities2Video task, we first sample 10 data points from Opens2v-nexus Yuan et al. (2025b). We then rewrite the original prompts into longer and noisier versions.

For the Video2Video task, in order to better approximate real-world scenarios, we divide it into three settings: *Story alignment:* Given a video, modify its style according to the prompt while keeping all other aspects unchanged. *Style alignment:* Given a video, modify the storyline according to the prompt while preserving the original video's style, characters, and semantics. *Semantic alignment:* Given a video, modify both the style and storyline according to the prompt while retaining the original characters and other semantic elements (e.g., generating a sequel to the video). For each task, we sampled 10 videos from SF20k Ghermi et al. (2024), then manually generate prompts for each video.

**Editing (Long Video).** We sampled 10 videos from SF20k Ghermi et al. (2024), then manually curated the editing prompt based on the content of the video.

**Segmentation (Long Video).** We randomly concatenated clips from DAVIS2017 Perazzi et al. (2016), resulting in 10 segmentation task instances that involve occlusions and cover diverse scenes.

**Agentic probing sets.** We include (1) a 50-instance storyboard→user-intent planning set to compare Single-Agent vs. Plan-Act, and (2) a set of standard pipeline tasks with expert references

Table 3: Key dimensions for analyzing video shots and editing.

| Category | Dimension |
|---|---|
| Intra-frame | 1. Shot Size
2. Shot Angle
3. Shot Location
4. Shot Subject
5. Shot Type (composition)
6. Shot Color (grading/tonality) |
| Intra-shot | 7. Shot Motion (camera movement)
8. Shot Speed |
| Inter-shot | 9. Cut Type
10. Transition |

to assess wPED, DepCov, and ReplanQ under injected failures. Memory analyses consider *trace memory* (historical trajectories), *user memory* (preferences), and *task memory* (e.g., storyboards).

## D.3 METRICS

### D.3.1 SUBJECT METRICS (TASK QUALITY).

**CLIP Score (command following).**  Measures text-video alignment between the user instruction (or storyboard-derived captions) and generated/edited outputs. We report the average CLIP similarity over sampled frames/clips; higher is better.

**DINO Score (subject consistency).**  Measures referential/identity stability by comparing DINO features between reference entities (images/key frames) and generated/edited frames; the higher the better.

**Segmentation: J/F/mIoU.**  We report region (**J**-mean, IoU) and boundary (**F**-mean) quality, as well as **J&F**-mean; higher is better.

**Understanding score.**  Normalized accuracy over curated long-video QA pairs that span both semantics and aesthetics (shot transitions, style, narrative).

### D.3.2 MLLM AS A JUDGE (PREFERENCE).

To complement subject metrics, we perform pairwise preference judgments using an open-source judge (e.g., **InternVL-3-78B**) and a closed-source judge (e.g., **Gemini-2.5-pro**). Judges are provided with the instructions, any relevant references, and debiased captions; preferences are aggregated via majority voting, with ties being discarded. We report average preference rates and include significance tests when applicable.

To ensure consistent and unbiased evaluation, we used a standardized prompt template for our MLLM judge. The template was designed to be comprehensive and force a structured output.

---

**MLLM Prompt**

**[System Role]**
You are a rigorous multi-modal video evaluation expert (MLLM as a judge). Based only on the provided frames/timestamps and text/control information, evaluate a single video with structured scoring and traceable evidence. Do not hallucinate unseen content.

**C1. Semantic Content Accuracy** (Objects & Scene)
- What to check: Are the specified object categories present and correct? Is the overall scene type (nature/city/indoor/outdoor/weather/terrain) correct and stable?

---

- Typical evidence: timestamps where required objects/scenes appear (or fail), brief notes on correctness.
- Anchors:
1: Objects/scenes largely wrong or missing; persistent mismatch in most segments.
2: Frequent mismatches; objects or scene type often incorrect or unstable.
3: Mostly correct but with noticeable lapses (e.g., brief wrong class or scene drift).
4: Correct and stable with only minor slips in a few moments.
5: Fully correct and stable throughout; no contradictory frames observed.

### C2. Multi-Object & Spatial Relations
- What to check: Correct object count, arrangement, occlusion, and relative relations (above/below, inside/outside, left/right, front/back) consistent with perspective.
- Typical evidence: frames showing relation satisfaction/violation (e.g., "cup above plate").
- Anchors:
1: Major errors in count/placement; relations frequently wrong or contradictory.
2: Multiple wrong relations or unstable layouts; occlusion frequently implausible.
3: Largely correct with occasional conflicts or transient misplacements.
4: Almost entirely correct; rare, minor inconsistencies.
5: Fully correct and stable; relations clear and consistently maintained.

### C3. Action / Behavior Accuracy (Human or Specified Agent)
- What to check: If the prompt specifies actions/poses ("running," "waving"), are they clear, continuous, and recognizable? If no action is specified, set null.
- Typical evidence: timestamps covering onset/continuity/completion of the action.
- Anchors:
1: Action absent or clearly wrong most of the time.
2: Frequent mismatches or fragmentation; hard to recognize the intended action.
3: Generally matches, but with noticeable distortions or brief interruptions.
4: Clear and continuous match, with minor imperfections only.
5: Strong, consistent match; clear start-to-end execution with no ambiguity.

### C4. Attribute Fidelity (Colors & Specified Attributes)
- What to check: Specified attribute values (color, pattern/material, key part attributes) are correct and temporally stable for the intended targets.
- Typical evidence: timestamps where attributes are accurate or drift (e.g., jacket color switches).
- Anchors:
1: Attributes largely wrong or unstable; frequent drift or contradictions.
2: Many errors or drifts; correctness not sustained over time.
3: Mostly correct with occasional small drifts or brief miscoloring.
4: Accurate and stable with rare, subtle deviations.
5: Fully accurate and stable across the evaluated span.

### C5. Style Consistency (Appearance & Cinematic Movement)
- What to check: (a) Visual/appearance style (oil painting, cyberpunk, monochrome) matches the prompt AND remains consistent; (b) Camera grammar/movements (zoom/-pan/dolly/tilt, etc.) match the prompt and remain consistent.
- Typical evidence: timestamps showing style adoption/drift; note which sub-aspect (appearance or camera) deviates.
- Anchors:
1: Style severely mismatched or mostly absent; camera grammar opposite or missing.
2: Frequent mismatches or drift in either appearance or camera style.
3: Generally matches with occasional drift or brief instability.
4: Clear and consistent match with only slight, rare issues.
5: Fully consistent in both appearance and camera grammar throughout.

> **C6. Overall Video–Text Consistency (set null if no text prompt)**
> - What to check: Holistic semantic alignment between video and text (theme, scene, actions, style coherence). This is a summary dimension; do not double-count fine-grained issues already noted above.
> - Typical evidence: timestamps representing core theme fulfillment or contradictions.
> - Anchors:
> 1: Largely mismatched; core theme or requirements not met.
> 2: Many inconsistencies across key elements (theme/scene/action/style).
> 3: Mostly correct with noticeable errors in secondary aspects.
> 4: Overall consistent with small mismatches that do not change the theme.
> 5: Highly consistent; strong semantic agreement with the text prompt.

### D.3.3  AGENTIC METRICS (PLANNING & RECOVERY).

To quantitatively evaluate the agent's planning capabilities, we designed three specialized metrics: Weighted Plan Edit Distance (wPED), Dependency Coverage (DepCov), and Re-planning Quality (ReplanQ). The precise definitions and calculation methods for these metrics are detailed below.

**wPED (Weighted Plan Edit Distance)**  wPED measures the structural similarity between the sequence of tool names in an agent-generated plan ($P_{pred}$) and an expert-authored reference plan ($P_{ref}$). The score is derived from the classic Levenshtein edit distance, denoted as $L(A, B)$, which calculates the minimum number of single-item edits (insertions, deletions, or substitutions) needed to transform sequence A into sequence B.

The wPED score is calculated by normalizing this distance and inverting the result, ensuring that a higher score indicates a better alignment. The formula is:

$$\text{wPED} = 1 - \frac{L(P_{pred}, P_{ref})}{\max(\text{len}(P_{pred}), \text{len}(P_{ref}))} \tag{1}$$

A higher wPED score (closer to 1.0) signifies a closer structural alignment to the expert plan.

**DepCov (Dependency Coverage)**  DepCov evaluates the logical correctness of a generated plan ($P_{pred}$) by measuring its adherence to a set of fundamental, rule-based dependencies inherent to video production workflows. Our evaluation is based on a predefined set of rules, such as the principle that content generation must precede content editing.

Let $D(P_{pred})$ be the set of all dependency pairs $(u, v)$ identified in the plan $P_{pred}$ according to our rules, where tool $u$ must appear before tool $v$. Let $D_{sat}(P_{pred}) \subseteq D(P_{pred})$ be the subset of those pairs where this ordering is correctly satisfied. DepCov is then the fraction of satisfied dependencies:

$$\text{DepCov} = \frac{|D_{sat}(P_{pred})|}{|D(P_{pred})|} \tag{2}$$

A higher DepCov score indicates that the agent's plan is more logically sound and respects the procedural constraints of the task.

**ReplanQ (Re-planning Quality)**  ReplanQ measures the agent's ability to efficiently and effectively recover from a simulated execution failure. The metric is designed to reward intelligent, minimal plan modifications.

Let $P_{orig}$ be the agent's initial plan, and let the failure occur at index $i$. The agent then generates a revised plan, $P_{replan}$. We compare the suffixes of both plans starting from the failure point, denoted as $P_{orig}[i :]$ and $P_{replan}[i :]$. ReplanQ is calculated using the same normalized Levenshtein distance as in wPED, applied only to these suffixes:

$$\text{ReplanQ} = 1 - \frac{L(P_{orig}[i :], P_{replan}[i :])}{\max(\text{len}(P_{orig}[i :]), \text{len}(P_{replan}[i :]))} \tag{3}$$

A higher ReplanQ score (closer to 1.0) indicates a more efficient and robust recovery, suggesting that fewer changes were required to correct the plan after the failure.

### D.3.4 REPORTING PROTOCOL.

For Generation/Editing, we report CLIP, DINO, and MLLM preference; for Segmentation, J/F/J&F; for Understanding, normalized QA accuracy. For agentic probing, we report wPED/DepCov/Re-planQ with and without memory (trace/user/task) and compare the Single-Agent framework with the Plan-Act framework.

## E DETAILED EXPERIMENT SETTINGS

### E.1 UNIVA'S CONFIGURATION

Plan Agent: Claude-sonnet-4, Act Agent: Claude-sonnet-4, Video Generation Model: Seedance-v4-480p, Video Understanding Model: InternVL3-38B, GPT-5, Video Editing: Runway Aleph, Video Segmentation: SAM-2, Image Generation Model:flux-kontext-pro

### E.2 BASELINE CONFIGURATIONS

**Generation** For all video generation tasks, we standardized the output resolution to 480p and a frame rate of 24 fps to ensure a fair comparison. For baselines that natively lacked support for multi-image or video-conditioned inputs, we implemented a standardized pre-processing pipeline to bridge the capability gap:

- For the Entities2Video task, where some baselines only accept a single image, we first merged the multiple input reference images into a single composite image. This composite was then used as the input.

- For the Video2Video task, for text-only baselines, we first employed a video captioning model (Qwen2-VL-72B) to generate a detailed description of the source video. This generated caption was then prepended to the user's instruction prompt to guide the generation process.

The specific baseline models were configured as follows:

- LTX-Video: We utilized the official model and followed the recommended settings provided in their public repository.
- Seedance: We used the seedance-v1-pro-t2v-480p and seedance-v1-pro-i2v-480p from Wavespeed API, consistent with our UniVA's generation module, to ensure a direct comparison of the agentic framework's contribution.
- Wan: We used the wan-2.2/t2v-480p and wan-2.2/i2v-480p also via the Wavespeed API.

**Understanding.** For all the understanding tasks, we are using a frame rate of 1 fps and a maximum of 128 frames.

**Editing.** For the video editing task, we use Runway Aleph as the baseline model. In the baseline pipeline, videos are clipped into 5-second clips and sent to the Aleph model with a task prompt. Then, the edited video clips are merged together for evaluation.

**Segmentation.** For the video segmentation task, we use Sa2Va-4B as the baseline model, we directly send the video into the baseline model together with the segmentation prompt.

## F ADDITIONAL QUALITATIVE CASE STUDIES

*Please generate a 30-second short documentary video based on the following story beats. 1. Close-up of clay meeting a spinning wheel; fingers press and a rib tool carves spirals as slip flicks outward under warm studio light. 2. Over-the-shoulder time-lapse: the vessel rises from cylinder to wide bowl; wet sheen glistens while the wheel slows. 3. Kiln-loading montage: …… line patterns emerge. 5. Morning reveal: final bowl on a wooden table beside steaming tea; the potter signs the foot and exhales in quiet satisfaction.*

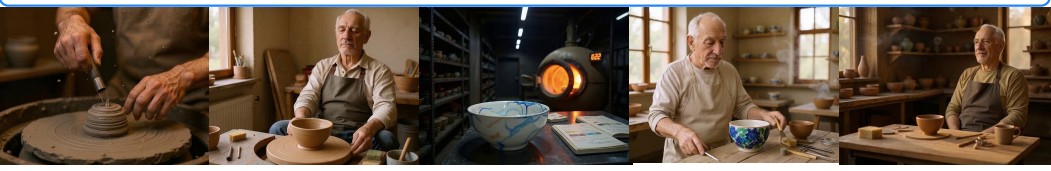

Figure 14: UniVA accurately generates the sequential process of pottery making, demonstrating strong temporal consistency and object persistence as the bowl evolves from clay to a finished product.

*0–3s: A man walks down a bustling city street at night, illuminated by vibrant neon lights and signs. He is dressed in a formal suit and tie, holding a smartphone, engrossed in its screen. The background features tall buildings with lit-up windows, creating a lively urban atmosphere. 3–7s: The camera slowly zooms in on the man's face, capturing his focused expression as he types on his phone. The neon lights reflect faintly on his glasses. …… 21–25s: The camera returns to the man, now seated on a bench in the park, still holding his phone but looking more relaxed. The city skyline is visible in the distance, blending the urban and natural elements. 25–30s: The camera slowly zooms out, showing the man in the peaceful park setting as the city awakens in the background, completing the visual and narrative transition.*

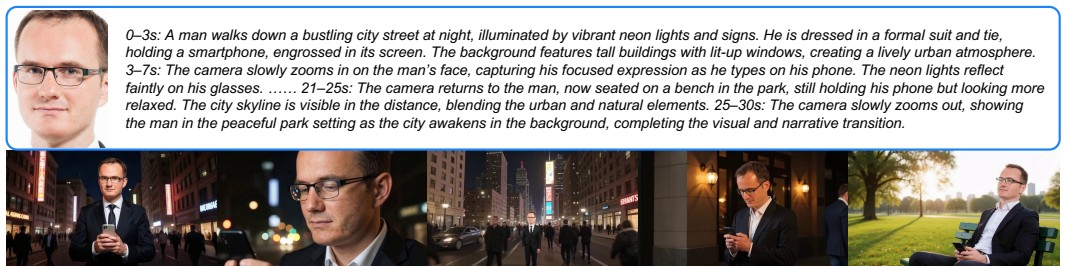

Figure 15: UniVA maintains the protagonist's identity flawlessly across drastically different scenes, lighting conditions (night vs. day), and camera angles, showcasing its advanced capability for robust, long-form character preservation.

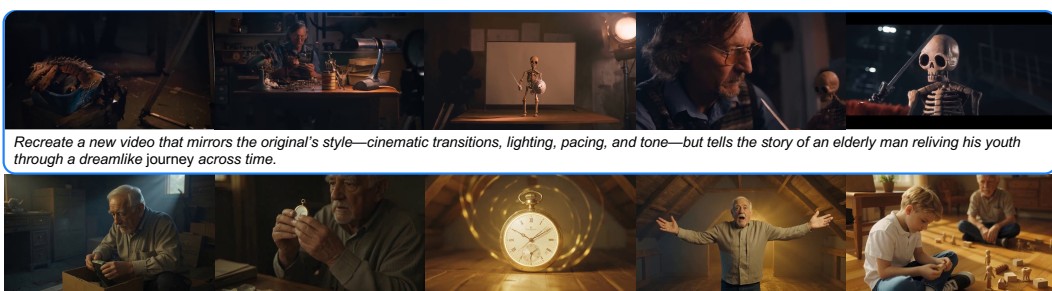

*Recreate a new video that mirrors the original's style—cinematic transitions, lighting, pacing, and tone—but tells the story of an elderly man reliving his youth through a dreamlike journey across time.*

Figure 16: UniVA interprets an abstract prompt to generate a complex narrative. It orchestrates a non-linear story arc, proving its capability as an intelligent storyteller powered by sophisticated planning.

*Please generate a 20-second advertising video based on the following product advertising requirements. 1. Kneading dough in hands, close-up shot, highlighting the texture of the dough. 2. Sprinkling cherry blossom petals on freshly baked bread, slow motion close-up. 3. Customers taste bread in the store and show satisfied smiles. 4. The brand logo appears, with the text: 'BreadTalk'.*

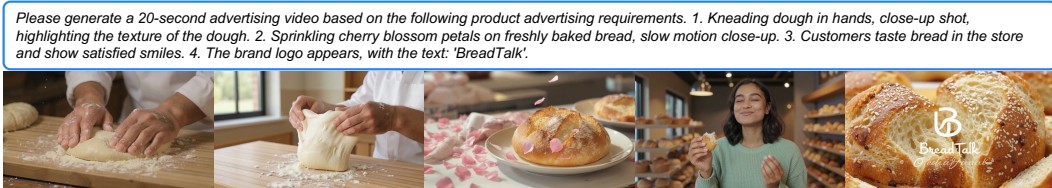

Figure 17: UniVA generates a coherent 20-second commercial that accurately follows the structured sequence of requirements—from kneading dough and showing customer reactions to applying the final brand logo.

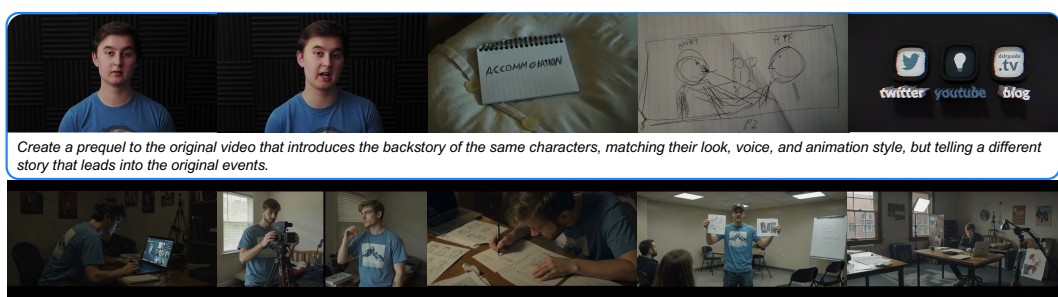

Figure 18: Given an original video, the agent not only maintains the original characters' style but also logically constructs a new backstory.

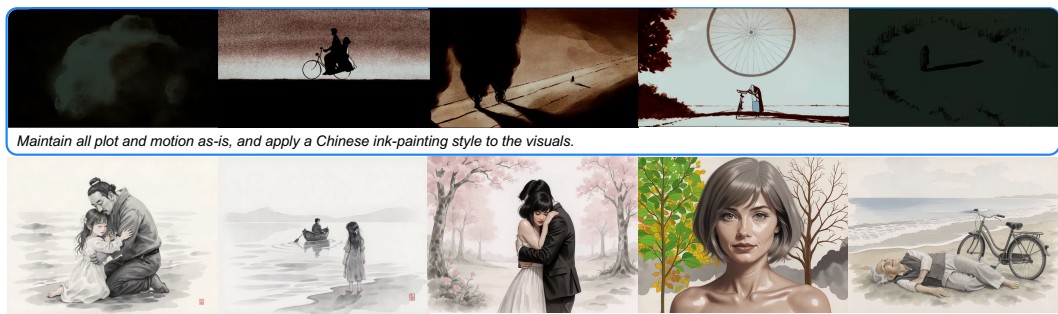

Figure 19: UniVA successfully applies a "Chinese ink-painting style" to the visuals while precisely maintaining the original video's plot, character motion, and scene composition.

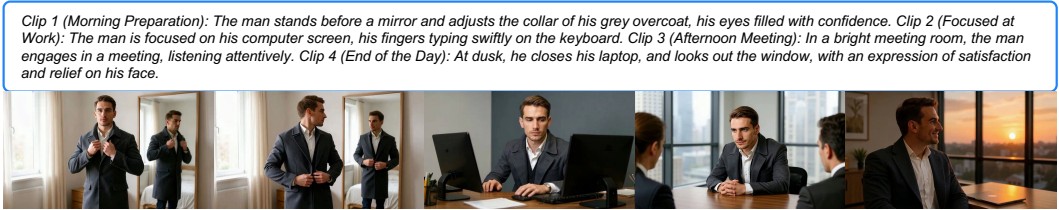

Figure 20: UniVA can well follow the user's long instructions and ensure consistency of characters in long videos.

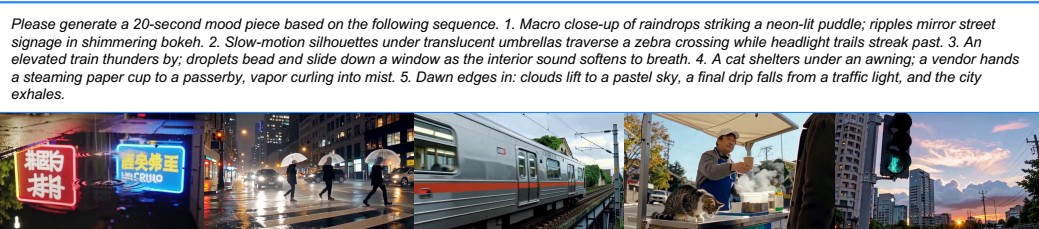

Figure 21: UniVA can understand and generate complex multi-camera scene transitions, producing multi-camera long videos.

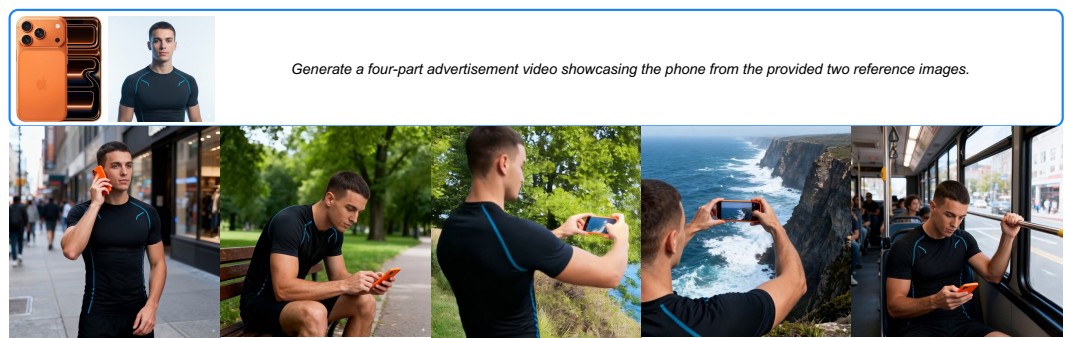

Figure 22: Univa can also maintain consistency well for multiple entity references.

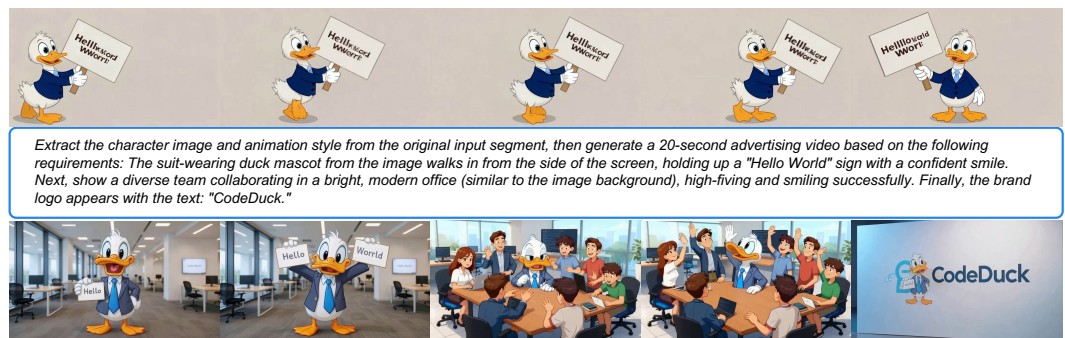

Figure 23: Univa can accurately analyze and understand the characters and style of a video, then seamlessly apply them to generate content.

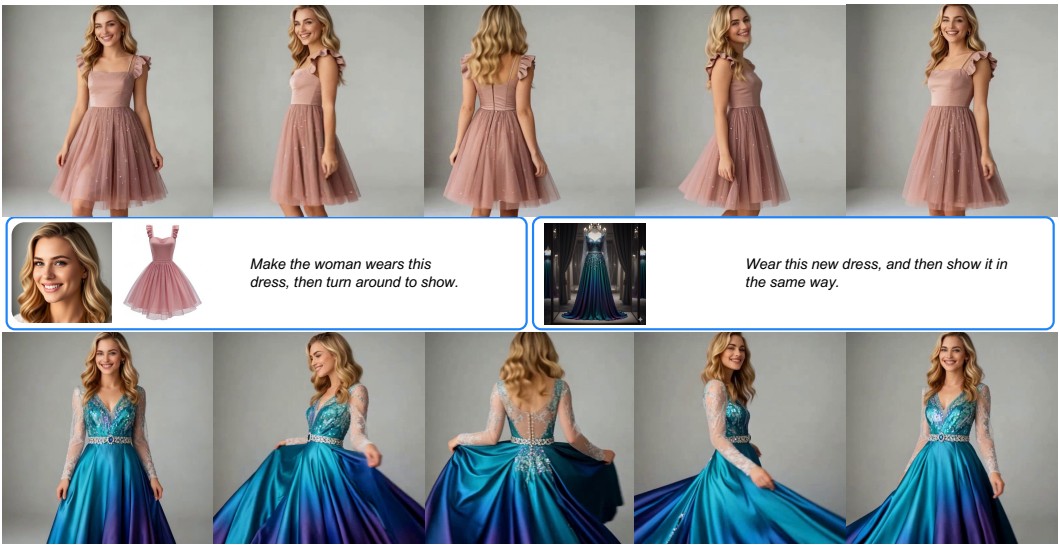

Figure 24: Univa can perform tasks through multi-turn dialogues by leveraging memory mechanisms and context.

