# OpenReview forum: "UniVA: Universal Video Agents towards Next-Generation Video Intelligence"
_ICLR.cc/2026/Conference — Submitted to ICLR 2026_

### Official Review · Reviewer_cun9 · 2025-10-19

**Soundness:** 2
**Presentation:** 2
**Contribution:** 2
**Rating:** 2
**Confidence:** 4

**Summary:**

The paper introduces a system called UniVA to help with complex video tasks. UniVA is designed to combine many different video tasks, such as understanding, cutting, editing, and creating videos, into one workflow. UniVA uses two types of AI agents to work. A planner agent receives a high-level request from a user and breaks it down into smaller, manageable steps. Then, executor agents take over to complete each of these individual tasks using various specialized tools. UniVA is built with a three-level memory design to handle long and complex tasks. The authors also created UniVA-Bench, a new tool for evaluating how well AI systems can handle complex, multi-step video assignments.

**Strengths:**

1. The paper is easy to understand. The motivation is clear.
2. The proposed system is complete. This needs substantial engineering efforts.

**Weaknesses:**

I am not an expert in designing video agent benchmarks, and I do not check the appendix. My criticism will focus on the technical parts.
1. This paper contains many over-claimed points, for example, next-generation video intelligence. I do not fully agree with the definition of next-generation video intelligence proposed in this paper.
2. Roughly speaking, the proposed agent works in a ReAct pattern. No significant planning contribution is presented. As for the memory part, the designed working memory is trivial and brings limited insights to the community.
3. As for the competitors, I am only familiar with the understanding part. Important baselines, such as VideoAgent and VideoAgent2, are missing. Only simple MLLMs are used for comparison, which is very unfair.
4. So, my point is that I cannot find truly new things brought to the video agent community with substantial experiments to prove them.

To some extent, HuggingGPT with a ReAct pattern could achieve the same effect as the proposed framework with some engineering efforts. I do appreciate the engineering efforts the authors have made, but the new insights are not enough for ICLR.

**Questions:**

See weaknesses.

---

> ### Author Response · Authors · 2025-11-23
> **1. Title**
>
> We sincerely accept the reviewer’s suggestion. Our original intention with the term "next-generation" was to highlight the paradigm shift from isolated, single-task models to unified, synergistic agentic workflows. However, we acknowledge this phrasing may appear over-claimed. We agree that the term "Next-generation Video Intelligence" can be subjective
>
> - To ensure scientific rigor and precision, we will modify the title in the final version to be more descriptive and grounded in our technical framework.
>
>     Original: UniVA: Universal Video Agents Towards Next-Generation Video Intelligence
>
>     Revised (Tentative): UniVA: Universal Video Agents Towards Open-Source Video Generalist

---

> ### Author Response · Authors · 2025-11-23
> **2. Insights to the community**
>
> First, we clarify a misconception: our agent adopts a plan-act architecture rather than a react architecture. In Figure 4, we compare these two architectures. Additionally, we argue that our work provides substantial insights and contributions to the video community as following:
>
> - First Open-Source Universal Platform: UniVA stands as the first open-source framework  capable of unifying video understanding, generation, editing, and segmentation into a single agent. By open-sourcing this platform, we aim to encourage the community to move beyond optimizing single-task models and focus on the immense potential of unifying diverse capabilities to solve complex, real-world workflows.
>
> - We demonstrate the critical value of Cross-Modal, Cross-Task Synergy. We show that, such as Understanding tools can actively enhance Generation and Segmentation quality, which is impossible for isolated models.
>
> - UniVA Benchmark: We address the lack of evaluation standards for this new field by introducing UniVA-Bench. Existing benchmarks predominantly focus on isolated, single-task performance (e.g., video quality or QA accuracy). UniVA-Bench fills this gap by shifting the evaluation focus to End-to-End Agentic Workflows.

---

> ### Author Response · Authors · 2025-11-23
> **3.Baselines**
>
> We appreciate the suggestion to compare with additional agent frameworks. While we recognize VideoAgent2 as a strong baseline, its codebase is currently not open-source, preventing us from conducting a fair and reproducible comparison. Instead, we have successfully completed comparisons with VideoAgent and VideoTree on the long video understanding task to address this concern.
> | Method | Accuracy |
> | --- | --- |
> | VideoAgent | 0.57 |
> | VideoTree | 0.73 |
> | UniVA | 0.76 |
>
> VideoAgent: Long-form Video Understanding with Large Language Model as Agent, ECCV 2024, https://arxiv.org/abs/2403.10517
>
> VideoTree: Adaptive Tree-based Video Representation for LLM Reasoning on Long Videos, CVPR 2025, https://arxiv.org/abs/2405.19209

---

> ### Author Response · Authors · 2025-11-27
>
> Dear Reviewer cun9,
>
> I hope this message finds you well.
>
> We have carefully addressed the questions and concerns you previously raised in our rebuttal. We would greatly appreciate any further feedback you could provide to ensure our responses have fully resolved your concerns.
>
> Your insights are highly valuable to us, and we remain fully available to clarify any remaining points. As the discussion deadline is approaching, we would appreciate hearing from you at your earliest convenience.
>
> Thank you specifically for the time and effort you have dedicated to reviewing our paper.
>
> Best regards,
>
> The Authors

---

### Official Review · Reviewer_YubW · 2025-10-29

**Soundness:** 3
**Presentation:** 3
**Contribution:** 3
**Rating:** 6
**Confidence:** 4

**Summary:**

The paper introduces UniVA, a multi-agent framework for unified video intelligence that integrates understanding, segmentation, editing, and generation. UniVA employs a Plan–Act dual-agent architecture, where a planner decomposes user goals into subtasks, and an executor carry out these steps using specialized modular tool servers. Besides, UniVA supports long-horizon reasoning and inter-agent communication while maintaining traceability of each action through a three-level memory system (global, task, user). To evaluate the framework, the authors introduce UniVA-Bench, a benchmark designed for multi-step video tasks spanning understanding, editing, segmentation, and generation. Experiments show that UniVA demonstrates competitive performance across a wide array of video tasks.

**Strengths:**

1.	The paper is well organized with clear diagrams and easy to follow.

2.	UniVA-Bench provides a systematic evaluation suite with new “agentic metrics” (wPED, DepCov, ReplanQ), filling a gap for multi-agent video systems.

3.	The experiments and visualizations are reasonable and well done.

**Weaknesses:**

1.	While the integration is strong, most modules (e.g., MCP protocol, planning agents, tool servers) are adaptations of existing frameworks rather than new technical inventions.

2.	While the ablation in Figures 6 and 7 shows improvements when incorporating user and task memory, the gains are relatively small.

3.	More ablations on user and task memory and comparisons with simpler orchestration baselines could be expanded to strengthen claims.

4.	The paper does not analyze latency, scalability, or hardware requirements, which are important for practical deployment.

**Questions:**

1.	How does the Planner dynamically adjust its plan when tool outputs deviate from expectations? Is there an explicit feedback or self-correction mechanism?

2.	How scalable is UniVA to hundreds of concurrent video tools or longer-than-minute sequences？Does the MCP protocol become a bottleneck?

3.	Could the authors provide more details on the computational efficiency of the end-to-end UniVA pipeline in practice?

---

> ### Author Response · Authors · 2025-11-23
> **1.Technical Inventions**
>
> We respectfully argue that the technical insight of UniVA should be viewed from a higher dimension: shifting from isolated task optimization to unified workflow automation. While "Plan-Act" and memory modules are established concepts in general LLM agents, our novelty lies in adapting these paradigms to solve the "Composite Workflow" problem in the video domain.
> - **First Open-Source Universal Video Platform**: Previous works largely focus on optimizing single tasks. For instance, a state-of-the-art generation model (e.g., LTX-Video ), it cannot "see" its own output to directly perform secondary creation (e.g., editing the background of the generated clip). These tasks currently rely heavily on human intervention to bridge the gap between understanding the output and formulating the next command. UniVA is the first open-source platform  to close this loop automatically. By unifying these capabilities, we provide a blueprint for the community to move beyond single-task optimization, demonstrating the immense value of an integrated system where generation is informed by understanding.
> - **Solving "Composite Workflows" via Agentic Synergy**: The core technical contribution is the realization of Cross-Modal, Cross-Task Synergy enabled by our architecture. We demonstrate that combining generalist reasoning with specialized tools yields performance superior to specialized models alone.
>     - As shown in Table 2(c), UniVA significantly outperforms the specialized baseline SA2VA on long video segmentation (J&F 0.2467 vs. 0.1524). UniVA dynamically leverages its Understanding module to resolve semantic ambiguities (e.g., distinguishing an object reappearance) that the segmentation tool cannot solve in isolation.

---

> ### Author Response · Authors · 2025-11-23
> **2.Ablation in Figures 6 and 7**
>
> We acknowledge that the absolute improvements in metrics like CLIP or DINO in Figures 6 and 7 appear modest. However, we argue that these metrics underestimate the structural and qualitative impact of the memory mechanism as follow:
>
> - Traditional metrics like CLIP/DINO focus on individual frames or short clips. However, the role of Task Memory (Storyboard) is to maintain narrative consistency across the entire video. As shown in Figure 7, the gain in the MLLM Judge score (which evaluates plot logic and continuity) is more pronounced than in CLIP scores. This confirms that memory is essential for understanding story and maintaining logical flow, a capability that is important for long-form video generation but traditional metrics show smaller shifts.

---

> ### Author Response · Authors · 2025-11-23
> **3.Ablations for Simpler Baselines**
>
> We believe our current experimental design already effectively isolates the contributions of our architecture compared to simpler baselines. We clarify our rationale below:
> -  Our ablation strategy was designed to test each memory module against the task most sensitive to it, rather than running all combinations. User Memory on Entities2Video: This task requires retrieving specific user assets like images, which is suitable for testing User Memory. Task Memory on LongText2Video: This task requires maintaining a long narrative structure, which is the specific function of the Storyboard in Task Memory.
> - And we show that the "Single-Agent" baseline presented in Figure 4 shows exactly as the "simpler orchestration baseline." This setup removes the specialized Plan-Act dual architecture and operates as a standard  ReAct-style agent. Our results show that this simpler orchestration is insufficient for complex video workflows. As shown in Figure 4, the Single-Agent baseline suffers a catastrophic drop in Success Rate (45.0% $\to$ 20.0%) and Plan Quality (wPED 0.117 $\to$ 0.050)

---

> ### Author Response · Authors · 2025-11-23
> **4.Latency & Scalability & Hardware**
>
> We thank the reviewer for this practical point. We will update a detailed imformation to the new version.
> ## Latency
> While total latency depends on the base models, for example the planning overhead introduced by UniVA (using claude) is typically <2s per step.
> ## Scalability
> Built on the Model Context Protocol (MCP), UniVA's tools are modular servers. This allows horizontal scaling where heavy tools (e.g., Video Gen) can be distributed across different GPUs or accessed via APIs.
> ## Hardware Requirements
> Due to the decoupled MCP design, UniVA does not require a massive cluster; users can run the core agent on a consumer CPU/GPU while offloading heavy model inference to remote servers or APIs, enabling low-resource deployment.

---

> ### Author Response · Authors · 2025-11-23
> **5.Dynamic Adjustment**
>
> UniVA incorporates an explicit self-correction mechanism within its iterative Plan-Act loop.
>
> - Our Planner and Actor operate in a continuous loop. After the Actor executes a tool, the Planner will check this intermediate result and evaluates it against the user's high-level intent before execute the next step.
>
> - And We explicitly evaluated this capability using the ReplanQ (Re-planning Quality) metric, which specifically measures the agent's robustness in detecting execution failures and generating valid recovery plans.

---

### Official Review · Reviewer_5atz · 2025-10-30

**Soundness:** 2
**Presentation:** 3
**Contribution:** 2
**Rating:** 6
**Confidence:** 3

**Summary:**

In this paper, the authors focus on the goal of creating a universal agent with broad-based video understanding and generation capabilities, sufficient for video creation workflow. To this end, the authors propose UniVA, a unified video agent system that incorporates separate planning and executing agents that interact with tool servers. The system also incorporates various forms of memory to ensure proper context. The authors evaluate their approach on various video understanding tasks, and they also present a novel benchmark named UniVA-Bench for future research.

**Strengths:**

1. The paper's goals are ambitious: A universal system for general video understanding and synthesis. They demonstrate a plausible system with promising results on some tasks.

2. The paper is well written and easy to understand.

3. The benchmark suite (UniVA-Bench) can be used for further research in the community.

**Weaknesses:**

1. The approach is very complex, and while some results on tasks are promising, there is still room for better performance given the high complexity of the approach.

2. It is unclear if there are any novel methods or models presented in the paper. This reads almost like a systems architecture paper that combines a number of prior models into one system.

3. More qualitative examples would be helpful in understanding the performance of the approach.

**Questions:**

1. Could the authors please elaborate on the contributions of the paper? Is this more of a systems architecture paper?

2. Would it be possible to include more qualitative examples in the paper?

3. Are there any ways the systems architecture could be simplified without sacrificing performance?

---

> ### Author Response · Authors · 2025-11-23
> **1.Complexity**
>
> We acknowledge the system's complexity. However, we view the current UniVA architecture as a balanced configuration optimized for "Universality", rather than an arbitrary accumulation of modules. We validated this balance through three dimensions:
>
> - Comparison with Simple Pipelines: To ensure a fair comparison on agentic tasks, we actually equipped single-model baselines with basic workflow wrappers (e.g., manually chaining a captioner with a video generator for Video2Video tasks). Even with these enhancements, Table 1 shows that UniVA significantly outperforms these simpler workflows (e.g., MLLM Judge: 4.068 vs. 2.123 for LTX-Video). This suggests that simple linear pipelines struggle to handle multi-constraint instructions.
>
> - Performance of Simplified Architectures: We explicitly explored a simplified version of our system—the "Single-Agent" architecture (without Plan-Act separation)—in Figure 4. While this simplified version is less complex, its Success Rate dropped drastically to 20.0% compared to 45.0% for the Plan-Act agent.
>
> - Moreover, since our framework is based on the MCP architecture, users can easily choose to unload certain tools. For example, if they do not wish to use the editing functionality, they can opt not to connect the editing tools.

---

> ### Author Response · Authors · 2025-11-23
> **2.Qualitative Examples**
>
> We have updated 3 pages additional qualitative comparisons in new version, Appendix F, showcasing scenarios where UniVA succeeds in maintaining narrative consistency where simpler baselines fail.

---

> ### Author Response · Authors · 2025-11-23
> **3.Contributions**
>
> We summarize our main contributions in the following three points:
>
> - First Open-Source Universal Platform: UniVA stands as the first open-source framework  capable of unifying video understanding, generation, editing, and segmentation into a single agent. By open-sourcing this platform, we aim to encourage the community to move beyond optimizing single-task models and focus on the immense potential of unifying diverse capabilities to solve complex, real-world workflows.
>
> - We demonstrate the critical value of Cross-Modal, Cross-Task Synergy. We show that, such as Understanding tools can actively enhance Generation and Segmentation quality, which is impossible for isolated models.
>
> - UniVA Benchmark: We address the lack of evaluation standards for this new field by introducing UniVA-Bench. Existing benchmarks predominantly focus on isolated, single-task performance (e.g., video quality or QA accuracy). UniVA-Bench fills this gap by shifting the evaluation focus to End-to-End Agentic Workflows.

---

### Official Review · Reviewer_27tr · 2025-11-01

**Soundness:** 2
**Presentation:** 2
**Contribution:** 2
**Rating:** 4
**Confidence:** 4

**Summary:**

This paper presents UniVA, a unified multi-agent framework designed to handle complex video tasks—including generation, understanding, editing, and segmentation—through a Plan-Act architecture enhanced with multi-level memory and modular tools integrated via the Model Context Protocol (MCP). The authors also introduce UniVA-Bench, a dedicated benchmark for evaluating multi-step video workflows.

**Strengths:**

- Built on the Model Context Protocol (MCP), UniVA supports a wide range of video, non-video, and non-AI tools in a modular and plug-and-play manner, enabling flexible and extensible task execution.
- The framework demonstrates comprehensive capabilities across diverse video-related tasks, integrating multiple functionalities into a unified pipeline.

**Weaknesses:**

- The paper's core technical components—the Plan-Act architecture and multi-level memory mechanism—are based on well-established paradigms in the agent literature, while the Planner itself is implemented using an existing LLM framework, thus providing limited novel technical insight specifically tailored to video intelligence.
- Evaluations in the benchmark primarily compare against non-agentic models, failing to adequately highlight UniVA’s advantages over recent video-specific agent systems.
- The small scale of UniVA-Bench—using only 10 videos per task—may undermine the generalizability and robustness of the experimental conclusions.
- There is a lack of in-depth analysis on failure modes, such as planning errors, tool invocation conflicts, or memory retrieval issues.

**Questions:**

- How does UniVA perform under real-time or low-resource conditions, and what are the computational requirements for each module?
- How does the Planner handle ambiguous or underspecified user instructions, and what mechanisms are in place for recovery?
- Can UniVA support interactive video tasks, and if so, how is dynamic user input incorporated during execution?

---

> ### Author Response · Authors · 2025-11-23
> **1.Novel Technical Insight**
>
> We respectfully argue that the technical insight of UniVA should be viewed from a higher dimension: shifting from isolated task optimization to unified workflow automation. While "Plan-Act" and memory modules are established concepts in general LLM agents, our novelty lies in adapting these paradigms to solve the "Composite Workflow" problem in the video domain.
> - **First Open-Source Universal Video Platform**: Previous works largely focus on optimizing single tasks. For instance, a state-of-the-art generation model (e.g., LTX-Video ), it cannot "see" its own output to directly perform secondary creation (e.g., editing the background of the generated clip). These tasks currently rely heavily on human intervention to bridge the gap between understanding the output and formulating the next command. UniVA is the first open-source platform  to close this loop automatically. By unifying these capabilities, we provide a blueprint for the community to move beyond single-task optimization, demonstrating the immense value of an integrated system where generation is informed by understanding.
> - **Solving "Composite Workflows" via Agentic Synergy**: The core technical contribution is the realization of Cross-Modal, Cross-Task Synergy enabled by our architecture. We demonstrate that combining generalist reasoning with specialized tools yields performance superior to specialized models alone.
>     - As shown in Table 2(c), UniVA significantly outperforms the specialized baseline SA2VA on long video segmentation (J&F 0.2467 vs. 0.1524). UniVA dynamically leverages its Understanding module to resolve semantic ambiguities (e.g., distinguishing an object reappearance) that the segmentation tool cannot solve in isolation.

---

> ### Author Response · Authors · 2025-11-23
> **2.Baselines & Fairness**
>
> We clarify two key points:
> 1. **Wrapper for Single Models:** Since most of our tasks are agent-centric (multi-step), "raw" single models cannot perform them directly. To ensure fairness, we constructed basic workflows for all single-model baselines to enable them to handle these tasks. For example, in Table 1, for the three base generation models in video-to-video task, we built a simple workflow for each model, starting with a video understanding module to comprehend the video, then passing the video caption and text prompt to the base models to enable them to complete the task.
> 2. **Agent Baselines:** We acknowledge the suggestion to compare with other agent. In the revised version, we will add comparisons with relevant agents to provide a more comprehensive and fair comparison. Now we have completed the comparison of VideoAgent and VideoTree (both good at long video understanding) on the long video understanding task. The results for the video generation task will be added later.
> | Method | Accuracy |
> | --- | --- |
> | VideoAgent | 0.57 |
> | VideoTree | 0.73 |
> | UniVA | 0.76 |
>
> VideoAgent: Long-form Video Understanding with Large Language Model as Agent, ECCV 2024, https://arxiv.org/abs/2403.10517
>
> VideoTree: Adaptive Tree-based Video Representation for LLM Reasoning on Long Videos, CVPR 2025, https://arxiv.org/abs/2405.19209

---

> ### Author Response · Authors · 2025-11-23
> **3.Benchmark Scale**
>
> We argue that the value of UniVA-Bench lies in the complexity and density of workflows rather than merely the count of source videos. Unlike traditional benchmarks where "1 sample = 1 model inference," a single task instance in UniVA-Bench triggers a long-horizon chain of dependent sub-tasks.
> - Taking the LongText2Video task as a example:  The task requires generating a coherent 60-second video. The Workflow Density: Since most state-of-the-art base generation models are limited to generating short clips (approx. 5 seconds), completing a single benchmark instance requires a minimum of 12 sequential generation steps (60s / 5s).
> - Beyond mere generation, the Planner must dynamically intervene. 1. Storyboard Planning: Decomposing the narrative into scene-by-scene beats before generation begins. 2. Consistency Management: Using previous clips as conditions for subsequent steps to maintain character/style consistency. 3. Final Composition: Merging clips and adding audio/subtitles.
>
> Consequently, a single "data point" in our benchmark actually evaluates multi turns, which providing a depth of evaluation that simple, large-scale benchmarks cannot offer.

---

> ### Author Response · Authors · 2025-11-23
> **4.Analysis of Failure**
>
> We acknowledge the need for deeper analysis. Our paper currently addresses planning failures through some metrics like ReplanQ metric, wPED (Fig. 3 ~ 5), specifically measuring the agent's ability to recover from execution errors. As shown in Figure 5, we analyzed specific failure cases where the agent without Global Memory failed to generate valid plans.
>
> In the revision, we will add a dedicated "Error Analysis" section in the Appendix.

---

> ### Author Response · Authors · 2025-11-23
> **5.Computational Requirements & Handling Ambiguity & Interactive Capabilities**
>
> ## Computational Requirements
> - UniVA is built on the Model Context Protocol (MCP), which decouples the system components. This allows the Planner/Executor and the Tool Pool to be flexibly deployed or changed.
> - For low-resource settings, users can easily configure UniVA to use API-based models (e.g., Claude/GPT-4o for planning, commercial APIs for generation) instead of heavy local GPUs. If there are sufficient GPU resourcesor, it also can use some sota open-source models (e.g., Qwen, Wan) running on local GPUs.
>
> ## Handling Ambiguity
> If an instruction is underspecified, the Planner won't generate a random plan. Instead it will ask user to input more requirements or retrieves successful trajectories from Global Memory to clarify intent before execution.
>
> ## Interactive Capabilities
> UniVA natively supports interactive tasks.
> - Mechanism: As illustrated in the "Multi-round Task" in Figure 11 (Right), the system incorporates dynamic user input (e.g., "Change the background...") as a state update.
> * Implementation: The Task Memory persists the context (e.g., segmentation masks, previous video clips) , allowing the Planner to treat new input not as a restart, but as a modification request applied to the existing context.

---

> ### Author Response · Authors · 2025-11-27
>
> Dear Reviewer 27tr,
>
> I hope this message finds you well.
>
> We have carefully addressed the questions and concerns you previously raised in our rebuttal. We would greatly appreciate any further feedback you could provide to ensure our responses have fully resolved your concerns.
>
> Your insights are highly valuable to us, and we remain fully available to clarify any remaining points. As the discussion deadline is approaching, we would appreciate hearing from you at your earliest convenience.
>
> Thank you specifically for the time and effort you have dedicated to reviewing our paper.
>
> Best regards,
>
> The Authors

---

### Author Response · Authors · 2025-11-23
**General Response**

Dear PCs, SACs, ACs, and Reviewers,

We sincerely appreciate the reviewers’ time and constructive feedback on UniVA. We are truly grateful for the insightful suggestions, particularly regarding the fairness of baselines and our system's contribution. In response, we have clarified our experimental setup to address these points and expanded the discussion on the system's efficiency and architectural novelty. We have also updated the manuscript as requested. We look forward to any further comments from the reviewers.

Best regards, The Authors

---

### Meta-Review · Area_Chair_jPEE · 2026-01-06

**Summary:**

This paper proposes an agentic framework for solving video understanding tasks, consisting of an LLM system where a high-level planner decomposes the query into a set of subgoals, where several specialized LLM actors call specialized tools to solve the specified task. The paper further proposes to have several memory systems: a global memory system shared across all agents, a task memory system for things relevant for the current task, and a user memory system describing the intents of the user.

Reviewers had several major concerns about the paper. First, the method proposed in the paper is rather standard, following many existing frameworks that consist of a high-level planner and lower-level models. In addition, the system is very complex, and there is a substantial amount of task-specific engineering for each task the system wishes to solve. Finally, the evaluation is very limited, consisting only of 10 examples for each task considered, with insufficient comparisons to existing agentic frameworks (all methods that are compared to are end-to-end neural networks).

**Reviewer Concerns:**

Reviewers had several major concerns about the paper. First, the method proposed in the paper is rather standard, following many existing frameworks that consist of a high-level planner and lower-level models. In addition, the system is very complex, and there is a substantial amount of task-specific engineering for each task the system wishes to solve. Finally, the evaluation is very limited, consisting only of 10 examples for each task considered, with insufficient comparisons to existing agentic frameworks (all methods that are compared to are end-to-end neural networks). While the complaint about the comparison with baselines is mostly resolved (the authors added several additional benchmarks to compare with), each of the remaining concerns, especially about novelty, is not well addressed. One direction the authors may think about improving novelty in this direction is to explore more closed loop feedback between individual components in the approach.

**Reviewer Scores:**

I believe Reviewer 27tr,  5atz, cun9  would maintain their scores, as the fundamental issues with the paper have not been resolved.

---

### Decision · Program_Chairs · 2026-01-26

Reject